# Exertional Exhaustion (Post-Exertional Malaise, PEM) Evaluated by the Effects of Exercise on Cerebrospinal Fluid Metabolomics–Lipidomics and Serine Pathway in Myalgic Encephalomyelitis/Chronic Fatigue Syndrome

**DOI:** 10.3390/ijms26031282

**Published:** 2025-02-01

**Authors:** James N. Baraniuk

**Affiliations:** Department of Medicine and Interdisciplinary Program in Neuroscience, Georgetown University Medical Center, 3900 Reservoir Rd NW, Washington, DC 20007, USA; baraniuj@georgetown.edu

**Keywords:** myalgic encephalomyelitis/chronic fatigue syndrome, cerebrospinal fluid, metabolomics, lipidomics, exercise, serine pathway, post-exertional malaise, PEM

## Abstract

Post-exertional malaise (PEM) is a defining condition of myalgic encephalomyelitis (ME/CFS). The concept requires that a provocation causes disabling limitation of cognitive and functional effort (“fatigue”) that does not respond to rest. Cerebrospinal fluid was examined as a proxy for brain metabolite and lipid flux and to provide objective evidence of pathophysiological dysfunction. Two cohorts of ME/CFS and sedentary control subjects had lumbar punctures at baseline (non-exercise) or after submaximal exercise (post-exercise). Cerebrospinal fluid metabolites and lipids were quantified by targeted Biocrates mass spectrometry methods. Significant differences between ME/CFS and control, non-exercise vs. post-exercise, and by gender were examined by multivariate general linear regression and Bayesian regression methods. Differences were found at baseline between ME/CFS and control groups indicating disease-related pathologies, and between non-exercise and post-exercise groups implicating PEM-related pathologies. A new, novel finding was elevated serine and its derivatives sarcosine and phospholipids with a decrease in 5-methyltetrahydrofolate (5MTHF), which suggests general dysfunction of folate and one-carbon metabolism in ME/CFS. Exercise led to consumption of lipids in ME/CFS and controls while metabolites were consumed in ME/CFS but generated in controls. In general, the frequentist and Bayesian analyses generated complementary but not identical sets of analytes that matched the metabolic modules and pathway analysis. Cerebrospinal fluid is unique because it samples the choroid plexus, brain interstitial fluid, and cells of the brain parenchyma. The quantitative outcomes were placed into the context of the cell danger response hypothesis to explain shifts in serine and phospholipid synthesis; folate and one-carbon metabolism that affect sarcosine, creatine, purines, and thymidylate; aromatic and anaplerotic amino acids; glucose, TCA cycle, trans-aconitate, and coenzyme A in energy metabolism; and vitamin activities that may be altered by exertion. The metabolic and phospholipid profiles suggest the additional hypothesis that white matter dysfunction may contribute to the cognitive dysfunction in ME/CFS.

## 1. Introduction

Myalgic encephalomyelitis/chronic fatigue syndrome (ME/CFS) is a chronic disease with disability, fatigue, post-exertional malaise (PEM), cognitive lapses, non-refreshing sleep, interoceptive distress, pain, and orthostatic complaints [1,2,3,4]. ME/CFS affects all age, sex, and racial and ethnic groups and costs the U.S. economy about USD 18–51 billion annually [5,6,7,8]. Prevalence is 0.2% to 1.3% [9]. Females are affected with a 2:1 to 4:1 prevalence. Recovery over time is poor, with only 0% to 8% having full recovery and 17% to 64% some limited improvement [10]. There are no objective diagnostic tests or FDA-approved drug therapies. There is extensive overlap between ME/CFS and Long COVID [11].

Post-exertional malaise (PEM) is the severe, prolonged disabling exacerbation of symptoms that follows small increments in cognitive, emotional, or physical exertion. The onset may be delayed by hours. Unlike controls, sleep is not refreshing, and rest does not lead to recovery of function. The exertional exhaustion can be prolonged for days and subjects may become bed-bound or inactive because of fatigue, pain, or cognitive inability to plan activities. PEM is a required, discriminating diagnostic criterion for ME/CFS [2,3,4]. The subjective and often uniquely personal nature of PEM makes it challenging to define and quantify [12,13]. Therefore, it is logical to objectively assess the consequences of an exercise provocation to understand mechanisms of exertional exhaustion.

This manuscript examined metabolomic changes in cerebrospinal fluid on the assumption that this fluid provides insights into brain physiology, ME/CFS pathophysiology, and the effects of exercise. To evaluate PEM, cerebrospinal fluid metabolites and lipids were contrasted in two independent cohorts of subjects who had lumbar puncture without exercise (non-exercise group) or after the second of two bouts of submaximal exercise performed on two consecutive days (post-exercise) [14,15,16,17]. Outcomes were analyzed by frequentist and Bayesian methods to contrast these statistical approaches. The sources of cerebrospinal fluid in plasma, blood–brain barrier, and brain cells were reviewed to highlight metabolic pathways that may be impacted by ME/CFS and exercise. The cerebrospinal fluid findings were put into context by comparison to previous studies of cerebrospinal fluid, plasma, urine, and fecal microbiome at baseline.

Frequentist multivariate and Bayesian linear regression analyses [18] were performed to assess the effects of disease (ME/CFS vs. sedentary control, SC), gender, exercise (non-exercise vs. post-exercise cohorts), age, and body mass index (BMI). The significant outcomes of the two methods were contrasted because they provide generally similar trends but with different interpretations [19,20,21]. The frequentist method-calculated point estimates to determine if there were significant differences (*p* < 0.05 by Sidak procedure to correct for multiple comparisons) between ME/CFS and SC groups as well as the other variables; 95% confidence intervals predict that 95% of future identical experiments will have results that fall within this interval. The Bayesian approach is designed to test if an investigation can improve the probability of finding a significant differences between groups. The approach begins by identifying the current state of knowledge or prior probability of a difference between groups. In this case, there was no a priori information about metabolic differences between ME/CFS and SC in cerebrospinal fluid using Biocrates mass spectrometry. Therefore, the default a priori assumption was that metabolite abundances in ME/CFS and SC were equivalent. Statistical tests for equivalence and linear regression were then applied to calculate the significance between the test groups; this provided the posterior probabilities for each metabolite. Comparison of the prior and posterior probabilities determined if the investigation significantly improved the ability to differentiate between ME/CFS and SC. Significance was defined by 95% credible intervals that do not include the null result (0 if results for the two groups are equivalent). Credible intervals indicate that there is a probability of 0.95 that the percentage of the results will match the hypothesis that ME/CFS and SC are different [22]. The valence will determine if the probability is in favor of ME/CFS (95% credible intervals > 0) or SC (95% credible interval < 0). In this way, the current data provided evidence to shift from the prior distribution (in this case equivalence) to the newly calculated posterior distribution (possible significant differences for metabolites and independent variables). Parallel computations of the frequentist and Bayesian methods allowed for comparison of the methods because the application of Bayesian methods to metabolomics is novel [18].

### 1.1. Cerebrospinal Fluid

Cerebrospinal fluid was studied because of the many central symptoms and magnetic resonance imaging [15,17,23,24,25] findings and the potential to provide insights into brain metabolism and pathophysiology of ME/CFS. This accessible fluid contains metabolites and mediators that are brain cell nutrients and end products of cellular metabolism. The brain parenchyma is compartmentalized away from the periphery and plasma by the tightly regulated choroid plexus, endothelial, pericyte, and astrocyte barriers (Figure 1).

The blood–cerebrospinal fluid barrier is formed by fenestrated capillaries that deliver plasma to the choroid plexus. Choroid plexus epithelial cells form tight junctions and actively transport selected nutrients into cerebrospinal fluid. Secretion is under tight circadian control [26,27]. The fluid is resorbed via arachnoid granulations. Arterial and arteriolar vessels penetrate into the brain parenchyma within a sheath of cerebrospinal fluid and astrocytic end feet [28]. Nutrients in the fluid are transported across the astrocyte barrier and exchanged via the brain interstitial fluid [29] with neurons, microglia, and oligodendrocytes. The interstitial fluid drains via the glymphatic flow into the venous system. Periventricular organs such as the median eminence have a hybrid of tight and fenestrated barriers [30]. The cerebrospinal and brain interstitial fluids may be able to exchange compounds with each other. The cerebrospinal fluid drains through arachnoid granulations into the venous blood. Factors that control the flux of metabolites, lipids, and other mediators through this plumbing system determine the composition of the cerebrospinal fluid. Insights into disease pathogenesis were inferred from differences in metabolite concentrations between ME/CFS and controls caused by the release of metabolites from brain cells and potentially myelin sheathes into the brain interstitial fluid with diffusion into the cerebrospinal fluid in combination with facilitated transport of plasma metabolites and lipids across the choroid plexus epithelial barrier. Changes induced by exercise and measured during the recovery period were portrayed as relative consumption if levels were higher in the non-exercise than post-exercise state, or increased production or accumulation if concentrations were higher after exercise. We predicted that the metabolite and lipid profiles in ME/CFS would reflect previous findings in cerebrospinal fluid, plasma, and urine, and the effects of exercise on cerebrospinal fluid.

### 1.2. Context for ME/CFS Metabolomics

There are two previous reports of cerebrospinal fluid metabolomics and one of exercise effects in control subjects. Walitt et al. [31] reported a hypometabolic profile in ME/CFS compared to control. Metabolites in tryptophan, glutamate, butyrate, polyamine, and tricarboxylic acid (TCA) cycle pathways were decreased in ME/CFS. Lipkin and Fiehn submitted Metabolomics Workbench Project PR000631, finding equivalence between ME/CFS and control except for reduced mannose and acetylcarnitine in ME/CFS [32]. The effects of exercise on cerebrospinal fluid were studied in control subjects who had lumbar puncture 1 h after running for 90 min [33]. Post-exercise levels were increased for lactate, purines, pyrimidines, dopamine, amino acid neurotransmitters, riboflavin, and nicotinamide ribose. The changes overwhelmingly implicate mitochondrial pathways.

The composition of cerebrospinal fluid in ME/CFS can be placed into the context of plasma, urine, and microbiome studies at baseline and after exercise. In general, ME/CFS subjects have lower plasma levels for mediators of glycolysis, oxidation reduction (redox), amino acid, urea, and lipid pathways than controls [34,35]. One hypothesis contends that pyruvate flux from glycolysis through the central mitochondrial enzyme pyruvate dehydrogenase (PDH) is blocked because of induction and expression of inhibitory PDH kinases [36]. This is based on disruption of the mechanisms of amino acid catabolism in glycolysis and the TCA cycle. There was no change for serine and glycine that enter glycolysis as three-carbon units to be metabolized to pyruvate. However, phenylalanine, tyrosine, leucine, and isoleucine that are metabolized into acetylCoA and anaplerotic amino acids such as glutamine and glutamate that enter the TCA cycle were decreased significantly, particularly in females with ME/CFS. 3-Methyl-histidine was elevated in ME/CFS men. These outcomes are consistent with previous findings of disruption of TCA, ketoacid, and amino acid metabolites [37,38,39,40,41]. Precursors and metabolites of purines, pyrimidines [42], ADP, and ATP [43] were reduced in ME/CFS. Nucleotides were rapidly liberated into plasma following exercise [33].

Naviaux et al. reported that ME/CFS had lower levels of plasma metabolites than controls, indicating a hypometabolic dauer-like state [44]. The dominant finding from their pathway analysis was that sphingolipid abnormalities constituted almost 50% of the metabolic disturbances associated with ME/CFS in both males and females [44]. Other phospholipid abnormalities added to the metabolic disturbances. Acyl cholines, androgenic, progestin, and corticosteroids were consistently reduced in plasma from ME/CFS. Ceramides were elevated in ME/CFS [45]. Acyl carnitine deficiency was reported in ME/CFS [46,47,48,49,50] although this was not a universal finding [45,48,51]. L-Carnitine supplementation has been reported to have beneficial effects on fatigue in ME/CFS [52,53].

Many plasma metabolites that are altered in ME/CFS are poorly annotated or may originate in the gut microbiome [44,45,54]. ME/CFS patients with disease duration <4 years [55] or with irritable bowel syndrome symptoms [56] have reduced microbial butyrate biosynthesis and a reduction in plasma butyrate, bile acids, and benzoate that became less marked with prolonged disease duration. Bacterial diversity was reduced [57] but fungi were unaffected [58]. The presence of microbial dysbiosis raises the hypothesis that the dysregulated products of gut microbes may have significant effects on cognitive, affective, perceptual, and peripheral functions in ME/CFS. The interactions remain to be determined.

Gender is an important variable in metabolomic studies with higher plasma levels of phenylalanine, glutamine, methionine, tyrosine, branched-chain amino acid, creatinine, and carnitines in men than women [59]. Males with ME/CFS had lower levels of sphingomyelins (*n* = 13), ceramide (*n* = 16), and other metabolites, while females had reduced ceramides (*n* = 18), phenyllactate, and AMP [44]. ME/CFS of both genders had reduced hydroxyisocaproic acid. Males had higher levels of serine, suggesting dysfunction of one-carbon metabolism in ME/CFS. Anaplerotic amino acids that can be metabolized through the TCA cycle were lower in females with ME/CFS [36]. Objective evidence of sexual dimorphism of ME/CFS is well documented [60,61,62] and will continue to influence efforts to understand mechanisms and develop treatments.

PEM has been studied in models of submaximal [14,15,16,17] and maximal cardiopulmonary exercise testing [63,64,65]. Metabolomic changes related to PEM must be placed into the context of normal responses. Exercise in control subjects has immediate, rapid effects on plasma that deliver metabolites and lipids to the choroid plexus and blood–brain/endothelial/parenchymal barriers. Acute aerobic exercise in control subjects induced four clusters of changes in metabolites, lipids, proteins, genes, and other mediators within the first hour postexercise [66]. Cluster 1 had elevated lactate, pyruvate, and malate, indicating anaerobic glycolysis with slowing in TCA cycle and accumulation of its intermediaries. Acyl carnitines, free fatty acids, phosphatidylcholine (PC), diacylglycerides (DAGs), cholesterol esters, ceramides, and sphingomyelins were elevated, which was interpreted as peroxisomal and mitochondrial metabolism of long-chain fatty acids with release of partially digested medium-chain acyl carnitines and fatty acids. Cluster 2 had elevated citrate, hypoxanthine, and xanthine from ATP turnover and urea from purine metabolism. Metabolites in Cluster 3 were decreased at 15 min but returned to normal at 1 h. Serine, glutamate, and tryptophan were in this group. Cluster 4 decreased through 1 h and beyond and included branched amino acids, phenylalanine, triacylglycerides (TAGs), and metabolites from the microbial biome.

The post-exercise state in ME/CFS has been associated with hypermetabolic plasma and elevation in the lactate-to-glucose ratio, acetate, urinary excretion of methylhistidine derived from post-exertional muscle protein digestion, and mannitol, which regulates intestinal barrier function but reduces levels of the purine metabolite hypoxanthine [67]. After maximal exercise, ME/CFS had elevation of plasma metabolites derived from glutamate metabolic pathways ranging from TCA to anaplerotic amino acid, butyrate metabolism, and the urea cycle [54]. Male ME/CFS subjects after exercise had the most alterations. Short-chain acyl carnitines were elevated in ME/CFS and controls. Many metabolites and lipids in ME/CFS were of probable microbiome origin.

Urine studies can provide insights because over 40% of metabolites have correlations with R > 0.4 between urine and plasma [68]. Urine metabolomics at baseline found no difference between ME/CFS and controls, which is at variance with studies that suggest hypometabolic or hypermetabolic states. After 24 h, controls showed significant elevation in many metabolites and lipids while ME/CFS had essentially no changes during recovery. Controls had elevations in steroids, acyl carnitines, and acyl glycines. Taurine, polyamine, and urea cycle intermediates were increased. Amino acid sub-pathways implicated branched-chain amino acids, tryptophan, cysteine, arginine, proline, methionine, and methylation. The absence of change in that ME/CFS cohort suggested an inability to respond to the exercise stressor.

This preamble implicates glycolysis, TCA cycle, diverse amino acid pathways, purine, pyrimidine, sphingolipid, and phospholipid pathways in ME/CFS pathology. Other pathways include cholesterol, pyrroline-5-carboxylate, riboflavin, peroxisomal, mitochondrial, and microbiome metabolism [44,45,68,69,70]. These pathways have intersections and share common metabolites that can be siphoned from one metabolic pathway to another. The metabolic modules were matched to KEGG (Kyoto Encyclopedia of Genes and Genomes) pathways then integrated into a metabolomic map.

Previous work identified significantly altered metabolites and lipids in ME/CFS cerebrospinal fluid [71]. Multivariate general linear models tested the frequentist hypothesis that point estimates for metabolites and lipids would have significant differences between ME/CFS and sedentary control (SC) subjects when corrected for gender, exercise, age, and body mass index (BMI). The null hypothesis was equivalence between ME/CFS and SC. The univariate contrast for disease (ME/CFS vs. SC) identified higher serine, sarcosine, sphingomyelins, and other phospholipids in ME/CFS than SC (Figure 2, Appendix A). The multivariate triple cross-product Disease x Gender x Exercise revealed that the non-exercise cohort had 25 metabolites and 26 lipids that were higher than SC for female and male subjects (ME/CFS > SC female non-exercise, ME/CFS > SC male non-exercise). SC had higher levels than ME/CFS for 11 metabolites and five lipids particularly for females postexercise (SC > ME/CFS female postexercise). The 42 metabolites and 33 lipids that were significantly different between ME/CFS and SC were tabulated, evaluated by KEGG pathways, and mapped to metabolic modules identified by the literature review (Figure 2). Metabolites that were higher in ME/CFS > SC are highlighted in yellow, and SC > ME/CFS in pink. An important caveat is that there is variability between metabolomics studies based on differences in the breadth of the metabolite and lipid panels, technical methods, gender, sample sizes, and clinical settings. Individual metabolites and lipids may not be consistently significant from one study to the next, but pathway analysis is felt to be more reproducible.

The purpose of this manuscript was to compare the lists of metabolites and lipids that were significantly different between ME/CFS and SC using Bayesian analyses compared to multivariate linear regression (Figure 2).

This pilot investigation used a hierarchical approach. Data were analyzed by Student’s *t*-test to demonstrate significant differences between ME/CFS and sedentary control (SC). This justified further evaluation by multivariate general linear modeling. Data were re-analyzed by Bayesian linear regression, Bayesian analysis for independent samples, and additional frequentist multivariate models to gain an appreciation of the yield of each statistical method. Results were displayed on the metabolic module map derived from the multivariate analysis in order to provide visual comparisons between ME/CFS and controls for each method and to contrast the effects of exercise and gender in both groups. The composite metabolic maps were examined to infer potential metabolic mechanisms of relevance to the pathophysiology of ME/CFS.

## 2. Results

Data were obtained for cohorts of non-exercise and post-exercise ME/CFS and sedentary control (SC) subjects (Table 1). A uniform protocol of recruitment, online and via telephone screening, was followed by in-person written informed consent, thorough medical history and physical examination for exclusionary conditions, supervision throughout study participation in the Clinical Research Unit, and follow-up after discharge.

ME/CFS had higher scores for the CFS questionnaire [72] and higher rates of fibromyalgia by 1990 pain and tenderness [73] and 2010 symptom criteria [74] (Table 2). Age and BMI were equivalent. Cerebrospinal fluid was clear, colorless, with normal total protein, albumin, IgG, erythrocytes, leukocytes, and differential counts (Table 3).

### 2.1. Student’s t-Test

As a first approximation, raw non-transformed non-exercise data were compared between ME/CFS and SC by *t*-test (uncorrected). Serine (*p* = 0.00094, Hedges’ g = 0.93) led a group of 16 metabolites that were significantly higher in ME/CFS than SC (Table 4). Three were elevated in SC more than ME/CFS. Dihydrofolate, 5-methyltetrahydrofolate (5MTHF), ratios of serine/5MTHF, and lipid levels were not different in this initial comparison. As a result, all further statistical studies used log10 transformed, normalized, and standardized data.

### 2.2. Multivariate Model (Figure 2)

Serine was identified as a novel central hub in cerebrospinal fluid metabolism in ME/CFS. Serine links glycolysis to glycine, one-carbon metabolism via 5-methyltetrahydrofolate (5MTHF), and phospholipid synthesis [75,76,77]. The inverse relationship between serine and 5MTHF indicated disruption of methylation, which was supported by the elevation in sarcosine (methylglycine), creatine, and creatinine and reduction in dimethylglycine and choline. Other metabolites affected by methylation were altered. 5-Methylthioadenosine, an intermediate in the methionine pathway, was elevated in ME/CFS. Purine and thymidylate synthesis and degradation were disrupted, which may be due to folate consumption or insufficiency. 1-Methyladenosine and 7-methylguanosine may be derived from epigenetic methylation of DNA and RNA followed by polymer digestion.

Serine is a precursor of phospholipid biosynthesis (Figure 2) [78]. It is the direct precursor of phosphatidylethanolamine (PE) and combines with palmitate to form ceramides and subsequently hexylceramides (HCERs). Serine is derived from 3-phosphoglycerate, which is the precursor of phosphoglycerides (PGs). Choline is formed downstream from glycine and sarcosine and is the precursor for phosphatidylcholines (PCs). ME/CFS had elevated numbers of PG (11 of 17 mass spectrometry targets, *p* = 0.000085 by two-tailed Fisher exact test vs. SC), PE (6 of 11, *p* = 0.012), sphingomyelins (6 of 14, *p* = 0.016), HCER (3 of 11), and PC (3 of 13).

Elevated glucose-1-phosphate, glucose-6-phosphate, citrate, and trans-aconitate indicated disrupted energy metabolism. Trans-aconitate may have short-circuited the TCA cycle in ME/CFS [79]. Citrate and glutamine were elevated, suggesting anaplerotic metabolism of TCA intermediates to amino acid backbones and other pathways [80]. The aromatic amino acids phenylalanine and tyrosine are metabolized to acetylCoA; disrupted CoA metabolism was suggested by increased pantothenate and cysteamine. Elevated dopamine in ME/CFS was unanticipated.

SC had elevated 5MTHF, riboflavin, and flavin monophosphate, suggesting a relative deficiency of these vitamins in ME/CFS, which would lead to disruption of enzymatic pathways where they are cofactors. 1-Methylhistidine, ornithine, proline, hydroxyisocaproic acid, N-acetylglutamine, and phenylacetylglutamine were elevated in SC, which added further evidence for amino acid imbalances in ME/CFS.

### 2.3. Bayesian Linear Regression

Bayesian linear regression assessed the relationships between analyte concentrations and the independent variables disease, gender, exercise, age, and BMI. Default priors were used. The hypothesis predicted that metabolites would shift the posterior probability distributions to support significant changes in the independent variables with 95% confidence intervals excluding 0. For disease, this meant finding either ME/CFS > SC or SC > ME/CFS after accounting for the other variables. Effects of metabolites on the other variables were also found. The null hypotheses predicted there would be no differences (ME/CFS = SC; male = female; non-exercise = post-exercise; no linear relationships for age and BMI).

The Bayesian and frequentist methods were both significant for the string of serine, 5MTHF, sarcosine, creatine, and creatinine as well as trans-aconitate and sphingomyelins (ME/CFS > SC) (Figure 3). Glucuronate was a unique addition with Bayesian regression. In agreement with the frequentist model, 5-methyltetrahydrofolate significantly predicted SC > ME/CFS and non-exercise > post-exercise status. Overall, ME/CFS status was predicted by sarcosine, trans-aconitate, glucuronate, 7-methylguanosine, serine, L-acetylcarnitine, and other metabolites and lipids (variables: ME/CFS, male, age, BMI; Table 5).

Additional relationships between metabolites and gender (Figure 4A), exercise cohort (Figure 4B), age (Figure 4C), and BMI (Figure 4D) were investigated. Male gender and non-exercise status were significant for fructose-6-phosphate, glucose-1-phosphate, and 5-thymidylic acid. Glutathione (reduced) was associated with male and post-exercise status. Male sex, age, and BMI variables were predictive with citric acid, glucose, lactate, and N-acetylalanine. Male sex and age were predictive for 2-aminoisobutyrate, thymidine, isocitrate, and allantoin. Male status was associated with elevated levels of hydroxyisocaproic acid and ornithine. Being in the post-exercise specimen group predicted higher levels of N-acetylglutamine and thiamine monophosphate. Asparagine, 2-aminobutyric acid, and dimethylglycine increased with age.

The multivariate general linear model (Figure 2) and Bayesian linear regression (Figure 3 and Figure 4) evaluated different elements of the point estimates and hypotheses and were found to have different lists of significant metabolites and lipids (Appendix A). However, the union and intersection of these significant findings were of interest for assessing the differences due to disease, gender, and exercise. For disease, the set included ME/CFS > SC for serine, sarcosine, creatine, creatinine, trans-aconitate, and five sphingomyelins. SC > ME/CFS had 5MTHF, 1-methylhistidine, and phenylglutamine (Table 6). Pathway enrichment using the pooled significant metabolites for SC > ME/CFS and ME/CFS > SC focused on pyrimidine metabolism and methionine metabolism, but also identified ammonia recycling, arginine and proline metabolism, spermidine and spermine biosynthesis, and glycine and serine metabolism (Table 7, Appendix A).

The effect of gender was evaluated by comparing the results of multivariate (Figure 2) and Bayesian linear regression (Appendix A, Appendix A). Gender was weighted towards males as they had 49 metabolites and eight lipids with levels that were higher than females. Females had two metabolites and nine lipids. The metabolites and lipids with significant changes were pooled for pathway enrichment, and those selected were Warburg effect, gluconeogenesis, glycolysis, pentose phosphate pathway, pyrimidine metabolism, glutamate metabolism, citric acid cycle, butyrate metabolism, pantothenate, and CoA biosynthesis (Appendix A).

Exercise caused the consumption of 13 metabolites, including 5MTHF, myoinositol, fructose-6-phosphate, glucose-1-phosphate, glucose-6-phosphate, and the pyrimidines dTDP and dTTP when the multivariate and Bayesian linear regression data (Figure 5) were compared. Conversely, 11 were higher in post-exercise cerebrospinal fluid, including glutathione (reduced), riboflavin, thiamine monophosphate, and butyrate metabolites. The pooled metabolites were enriched for glycolysis, gluconeogenesis, taurine, and hypotaurine metabolism (Appendix A). Lipids (*n* = 43) were elevated before exercise, including 14 phosphatidylglycerides, 11 phosphocholines, 3 phosphoethanolamine plasmalogens, and 3 sphingomyelins. Lipids were enriched for sphingolipid pathway and phospholipid biosynthesis (Appendix A).

### 2.4. Bayesian Analysis for Independent Samples

Bayesian analysis for independent samples and matching multivariate general linear models were used for direct comparisons of ME/CFS vs. SC in the non-exercise and post-exercise datasets, and for exercise effects in non-exercise vs. post-exercise cohorts for ME/CFS and SC. The Bayes analysis tested the hypothesis that metabolites were differentially expressed and would significantly differentiate between ME/CFS and SC (ME/CFS≠SC, ME/CFS > SC with posterior probability > 0 or SC > ME/CFS with posterior probability < 0) based on the 95% credible intervals that excluded 0. Age and BMI were covariates. The null hypothesis was that ME/CFS = SC and that the 95% credible intervals included 0. Separate analyses tested the differences between non-exercise and post-exercise data, and then the effects of exercise on ME/CFS and SC.

Analysis of ME/CFS vs. SC non-exercise identified elevated serine and palmitate in ME/CFS suggesting blockage of sphinginane synthesis (Figure 5A). Sarcosine, creatine, and creatinine that are derived from serine were also elevated. 5-Methylthioadenosine, an intermediary of methionine metabolism, and taurine, another sulfur-containing metabolite, were elevated. Purine metabolism was implicated by elevated hypoxanthine, xanthosine, and AMP as well as the degradation products 7-methylguanosine, 1-methyladenosine, and acadesine. Dihydroorotic acid and 3-ureidoproprionate from the pyrimidine pathway were elevated. Glucose-1-phosphate and trans-aconitate indicated alterations in glycolysis and TCA cycle, respectively. Dopamine, phenylalanine, lysine, and its metabolites aminoadipate and L-acetylcarnitine were elevated. Four phosphatidylglycerols were elevated. ME/CFS had excess FFA(22:0), tetradecanedioic acid, homocysteic acid, and 2,3-butanediol.

The only analyte that was higher in SC than ME/CFS before exercise was hydroxyisocaproic acid. Overall, 18 metabolites matched the multivariate outcome (44%).

The post-exercise condition continued to show elevated serine in ME/CFS > SC (Figure 5B). ME/CFS had higher abundances of serine, dihydroorotic acid, and three sphingomyelins. Conversely, after exercise, ME/CFS had reduced levels of 5-methyltetrahydrofolate, biotin, flavone, riboflavin, and glycerate compared to SC (SC > ME/CFS post-exercise), suggesting consumption of these cofactors in ME/CFS during exertion and recovery.

ME/CFS had reduced levels of amino acid metabolites, including phenylalanine, tyrosine, and 3-hydroxyanthranilic acid, a breakdown product of tryptophan (Figure 5B). For phenylalanine, this was the opposite of the non-exercise situation (Figure 5A). Other metabolites that were decreased in ME/CFS post-exercise were 2-hydroxy-3-methylbutyric acid, 1-methyl-L-histidine, xanthylic acid, uridine, and urea.

Exercise had a large impact on ME/CFS judging by the significant differences between the non-exercise and post-exercise data (Figure 6A). ME/CFS had 29 metabolites and 37 lipids that were higher before exercise than after exertion (ME/CFS non-exercise > post-exercise, yellow highlights). Serine levels were not affected by exercise, but the elevated ME/CFS non-exercise levels of the downstream intermediates 5MTHF, glycerate, O-acetylserine, creatinine, and phosphatidylethanolamines that rely on serine were decreased significantly after exertion, indicating net consumption or the cessation of manufacture. Metabolites of glycolysis were higher before exercise. The aromatic amino acids phenylalanine and tryptophan and their products dopamine and 5-hydroxytryptophan, respectively, decreased following exercise. Other metabolites that became depleted following exercise in ME/CFS were biotin, the NMDA receptor agonist homocysteic acid, lysine and its metabolite aminoadipic acid. Similarly, levels of phosphatidylglycerides, phosphatidylcholines, phosphatidylethanolamines, and sphingomyelins were higher in non-exercise samples. Exercise caused decreased synthesis or increased consumption of these compounds in ME/CFS.

The metabolites that rose after exercise (post-exercise > non-exericse) in ME/CFS were the reduced form of glutathione, acadesine, N-acetylglutamine, and thiamine.

Exercise in SC caused significant alteration in metabolite abundances (Figure 6B). Levels were higher before exercise (SC non-exercise > post-exercise, yellow highlighting) for 5MTHF, 2-isopropylmalic acid, and lipids. However, many more metabolites had higher abundances after exercise (SC post-exercise > non-exercise, pink highlighting). Increased glucose was available for glycolysis with elevated production of lactate and alanine from pyruvate. Tyrosine, phenylalanine, methionine, and its metabolite 2-aminobutyrate were increased. Metabolism through glycine was suggested by elevated sarcosine, creatine, and hippuric acid with a reciprocal decrease in 5MTHF that may indicate its consumption during exercise. Hippuric acid is a microbiome metabolite. The elevation of L-carnitine may indicate release as a result of fatty acid transport into peroxisomes and mitochondria. Riboflavin and thiamine monophosphate both increased but it is not clear why these vitamins would be diverted from cells into the cerebrospinal fluid. N-acetylglutamine may participate in acetyl transport across mitochondrial membranes. 2,3-Butanediol and 2-aminooctanoic acid may have been sourced from the microbiome.

The differences between ME/CFS (Figure 6A) and SC (Figure 6B) were visually striking and suggest consumption in ME/CFS (non-exercise > post-exercise, yellow highlights) and metabolite production and excretion into cerebrospinal fluid in SC (post-exercise > non-exercise, pink highlights).

### 2.5. Frequentist Multivariate General Linear Model

In order to complete the comparison of statistical methods, the frequentist multivariate analysis that used disease status as the independent factor and age and BMI as covariates (but without gender) was contrasted with Bayesian analyses of independent samples for ME/CFS vs. SC for the non-exercise cohort (Figure 5A and Figure 7A), ME/CFS vs. SC in post-exercise samples (Figure 5B and Figure 7B), and effects of non-exercise vs. postexercise on ME/CFS (Figure 6A and Figure 8A) and SC (Figure 6B and Figure 8B).

There was good agreement between statistical methods for non-exercise comparisons of ME/CFS vs. SC, as 19 metabolites in Figure 7A matched the 25 (76%) found by Bayesian analysis (Figure 5A). The string of serine, sarcosine, creatine, creatinine, and trans-aconitate, and metabolism of purines and pyrimidines were recapitulated. However, the post-exercise comparisons had fewer significant changes (Figure 7B). Only cysteamine was elevated in ME/CFS > SC. SC had higher levels than ME/CFS for phenylalanine, tyrosine, 1-methylhistidine, and riboflavin by both methods (Figure 5B and Figure 7B).

The multivariate analysis for the effect of exercise (without gender) (Figure 8A) found 31 significant metabolites that were elevated in ME/CFS non-exercise. There was 63% agreement with the Bayesian analysis (Figure 6A). However, the two methodologies led to opposite valences for acadesine, thiamine monophosphate, glutathione (reduced), and N-acetylglutamine, which may have been related to excluding gender in these comparisons. Exercise in SC produced higher levels of 3-hydroxybutanoate, which is a ketone body; aspartate derived from the TCA cycle; thiamine monophosphate; and riboflavin (post-exercise > non-exercise, Figure 8B). Only riboflavin and N-acetylglutamine matched the Bayesian analysis (Figure 6B). 

The metabolites and lipids that were significantly altered between ME/CFS and SC in the non-exercise and post-exercise cohorts by Bayesian and multivariate analysis (Figure 5 and Figure 7) were pooled and used for pathway enrichment. The comparison of non-exercise ME/CFS vs. SC shared 22 metabolites and lipids (58%) that were enriched for methionine metabolism, phenylalanine and tyrosine metabolism, pyrimidine metabolism, and glycine and serine metabolism (Appendix A). The post-exercise comparison shared only 5 (19%) and enrichment did not identify any significant pathways.

The effects of exercise in ME/CFS were determined from the non-exercise and post-exercise differences in Figure 6A and Figure 8A. There was 91% agreement between statistical methods for the 75 analytes, including fourteen phosphoglycerides, eight phosphocholines, five phosphoethanolamines, and four sphingomyelins. The metabolites were enriched for thiamine metabolism; biotin metabolism; ammonia recycling, glycolysis, and other sugar metabolism pathways; amino acid pathways; and Warburg effect (Appendix A).

The metabolites that were significantly altered by exercise in SC (Figure 6B and Figure 8B) were enriched for glycolysis, gluconeogenesis, and methionine metabolism pathways (Appendix A). Only 11 of the 57 analytes (19%) were shared between methods.

## 3. Discussion

The baseline non-exercise specimens compared ME/CFS versus the control state. There was good agreement between outcomes of the frequentist multivariate method (Figure 2) and Bayesian linear regression (Figure 3), and between ME/CFS and SC by multivariate (Figure 5A) and Bayesian (Figure 7A) analyses. Our ME/CFS cohort had a hypermetabolic profile with metabolite and lipid levels that were higher in ME/CFS than SC. The hypermetabolic profile was in contrast to the more hypometabolic findings reported in cerebrospinal fluid by Walitt et al. [31] and Lipkin and Fiehn [32] and the consensus findings from plasma reviewed in the Introduction. Naviaux et al. [44] placed their findings into the context of the cell danger response paradigm [81,82,83] and proposed that ME/CFS fit into a hypometabolic dauer-like state [84,85]. Our profile had more in common with the hypermetabolic post-exercise plasma in ME/CFS [67], post-exercise control subjects with metabolite and lipid clusters 1 and 2 [66], and the profiles in metabolic syndrome and responses to infection, inflammation, and environmental cell injury. 

The cell danger response hypothesis may be analogous to the unfolded protein response, endoplasmic reticulum stress response, mitochondrial unfolded protein response, and related mechanisms [86,87]. The ancient mechanism provides a unified mechanism to improve cell and host survival after viral attack or other injury. The acute cell danger response initiates a series of functional modifications or shifts [81] that were matched to the findings of the current study.

Metabolic shift: Cellular metabolism shifts from net polymer synthesis (transcription, translation) to monomer synthesis to prevent the hijacking and assembly of cellular resources by intracellular pathogens. The shift leads to increased levels of purines, pyrimidines, and amino acids, and the products of polymer degradation. This condition was detected in our metabolomics analysis. RNA and DNA digestion releases 7-methylguanosine and 1-methyladenosine. Acadesine is a catabolite of adenosine. Degradation of proteins releases N-acetylglutamine, N-acetylalanine, and other N-acetyl-amino acids from posttranslational N-terminal capping of proteins. Posttranslational methylation of proteins generates 1-methylhistidine, which was increased in SC. 3-Methylhistidine was elevated in ME/CFS urine [41] but there is confusion about nomenclature of these derivatives because of the numbering of the imidazole ring [88]. 3-Ureidoproprionate from pyrimidine catabolism was higher in ME/CFS than SC, especially in the non-exercise cohort. The elevation may have been due to blockages in purine and pyrimidine synthetic pathways and “back-up” of hypoxanthine, xanthine, orotic acid, and dihydroorotic acid in ME/CFS at baseline followed by subsequent consumption during exertion. Folate and methyl transfer reactions were implicated. Levels in SC became elevated as a result of exercise.

Methylation shift: Modifications of histone and DNA methylation will alter gene expression, leading to increased cytokine and other transcriptional programs. DNA methylation implicates biotin metabolism [89]. Biotin was elevated in ME/CFS prior to exercise. It is essential for mitochondrial carboxylases. Biotinylation of histones leads to gene silencing and modifies DNA repair and chromatin structure [90,91]. Tetrahydrobiopterin is elevated in serum of ME/CFS patients with orthostatic intolerance [92,93].

Methylation can be altered through folate, methionine, and glutathione mechanisms. The serine–glycine–5MTHF shuttle acts in mitochondria and the cytoplasm [75]. The methylation of glycine to produce sarcosine (Figure 2 and Figure 3) was found, as well as an apparently age-related blockage of methylation that would form dimethylglycine and choline (Figure 4C). An extension of this pathway leads to the synthesis of creatine from methionine and glycine which requires 5MTHF and thiamine (Figure 6 and Figure 8). 5-Methylthioadenosine, an intermediate in the methionine pathway, was elevated in ME/CFS. Under the oxidizing condition of the cell damage response, single carbon metabolism proceeds via 5MTHF for synthesis of thymidylate and purines, methionine and S-adenosyl-methionine for spermidine and spermine synthesis, and production of glutathione and its chemical reduction in redox reactions [81]. Increased expression of methylene tetrahydrofolate dehydrogenase 2-like (MTHFD2L) in mitochondria diverts 5MTHF away from S-adenosyl-methionine and DNA methylation towards the cytosol for synthesis of purines and 5-thymidylate (dTDP).

Serine was centrally placed in the metabolomic network (Figure 2) and linked glycolysis to glycine, 5MTHF, 1-carbon, and phospholipid metabolic pathways [75,76,77]. This pathway is dysfunctional in cancer and other syndromes and is known to short-circuit glycolytic intermediates towards formate in a manner similar to the Warburg pathway that diverts glycolytic intermediates to lactate instead of feeding the tricarboxylic acid (TCA) cycle [75]. Formate was not measured in our assay. Glycerate, an alternative breakdown product of serine that reenters glycolysis, was reduced in ME/CFS and may indicate decreased active metabolism of serine, diversion to other pathways, or consumption via glycolysis during exercise.

Phospholipid shift: Increasing the length and number of double bonds in long acyl chains of phospholipids stiffens the cell, endoplasmic reticulum, and mitochondrial membranes. This may interfere with pathogen envelopment and egress. Serine is intimately involved in lipid biosynthesis as the direct precursor of phosphatidylethanolamine (PE) and ceremide that forms by the combination of serine with palmitate [78]. Serine’s precursor, 3-phosphoglycerate, is the precursor of phosphoglycerides (PGs). The path from serine to glycine leads to sarcosine and choline synthesis and formation of phosphatidylcholines (PCs). ME/CFS had more phosphoglycerides (*p* = 0.000085 by two-tailed Fisher exact test vs. SC), phosphoethanolamines (*p* = 0.012), and sphingomyelins (*p* = 0.016) than SC (Figure 2, Appendix A). The sphingomyelins in ME/CFS had longer acyl chains (19.6 ± 0.7 vs. 16.5 ± 1.0, *p* = 0.036) and marginally more double bonds (3.0 ± 1.0 vs. 1.5 ± 0.5, not significant). The elevated sphingomyelins may indicate release from white matter myelin sheaths and injured oligodendrocytes [23,24,25]. Phospholipid synthesis is dysfunctional in ME/CFS [70]. It is not clear if these changes in lipids would be sufficient to alter lipid raft mechanics and membrane fluidity or reduce red blood cell deformability in ME/CFS [94,95].

Myelin is made up of approximately 40% cholesterol esters; 40% phospholipids, including ethanolamine plasmalogens; and 20% galactosylceramides, including sulfatides [96]. This composition is similar to lipid rafts. Myelin membranes of oligodendrocytes are enriched in ceramides, cholesterol, and oleic acid derivatives. Phosphocholines comprise about 7% of myelin lipid but have a faster turnover of days to weeks that may increase during white matter injury and repair. Serine is necessary for new synthesis of choline for phosphocholines and phosphoethanolamines. Astrocytes have the highest neutral lipid content and are the major site for mitochondrial fatty acid beta-oxidation in the brain. Lipid class and fatty acid profiles vary across the brain and correlate with structure (e.g., T1w/T2w MRI ratios), functional connectivity, and differential functions of synaptic membranes of inhibitory (GABA) and excitatory (glutamate) neurons [97]. The dynamic exchange of lipids between cerebrospinal fluid, different cell types, and myelin is largely unexplored, and it is unclear whether cerebrospinal fluid levels in ME/CFS reflect membrane or energy needs versus structural axon myelination.

Paracrine shift: Cell lysis releases constituents into the surrounding tissue. The intracellular compounds that are released may include neurotransmitters and hormones such as N-acetylglutamine, homocysteic acid, dopamine, 5-hydroxytryptophan, and ATP that warn neighboring cells and distant effector cells of the danger. The presence of increased AMP in the cerebrospinal fluid invokes the purinergic hypothesis [81]. The intracellular concentration of ATP is estimated to be 1 to 5 mM and is nearly one million times higher than in the extracellular environment (5–10 nM). When a cell becomes injured or lysed, it will release its ATP and other nucleotides, hormones, and metabolites into the surrounding area, which will generate an intense local paracrine signal to warn adjacent cells of the danger and the presence of a pathogen. The ATP acts rapidly as a neurotransmitter and over a longer period for trophic modulation of proliferation, differentiation, mobility and death [98]. Glutamine is the amino acid with the highest concentration in cerebrospinal fluid and acts trophically as a shuttle in the brain between oligodendrocytes, astrocytes, and neurons to provide substrate for glutamate and gamma-aminobutyric acid neurotransmitter, anaplerotic amino acid synthesis, and TCA cycle energy metabolism [99]. Other compounds released upon injury may act synergistically to intensify and broaden the warning signal.

Antimicrobial shift: Many of these compounds have antiviral and antimicrobial properties when released into the local environment.

Sickness behavior shift: ATP metabolites, dopamine, and other neurotransmitters participate in activation of arousal and autonomic systems in the dorsal midbrain and brainstem that are dysfunctional in ME/CFS [100,101,102,103,104,105]. Their effects may alter behavior in order to drastically reduce activity, lengthen sleep patterns, immobilize the site of bodily injury, conserve energy resources for tissue repair, and isolate individuals to prevent the spread of infection to others. Prolonged expression of these protective behaviors may be interpreted as fatigue, reduced motivation, apathy, anhedonia, or abulia.

Noncoding nucleic acid shift: Endogenous retroviruses [106], long interspersed nuclear elements (LINEs), and other mobile genetic elements become mobilized, which may generate novel genetic variations. Alteration of miRNA may fit in this category [16].

Mitochondrial shift: Increased autophagy and mitochondrial fission may serve to remove intracellular pathogens. From the metabolomic perspective, many of the central pathways that are dysregulated in ME/CFS involve mitochondria. Exercise caused elevation in cerebrospinal fluid levels of lactate, purines, pyrimidines, dopamine, amino acid neurotransmitters, riboflavin, and nicotinamide ribose [33]. The changes overwhelmingly implicated mitochondrial pathways. For example, TCA cycle, amino acid synthesis, and catabolism are localized to mitochondria. Proliferating B cells from ME/CFS subjects have lower mitochondrial mass [107]. When stimulated in vitro, B cells preferentially utilize essential amino acids compared to control. Mitochondrial dysfunction is present in ME/CFS, but only severely affected patients have both mitochondrial and glycolytic impairment [108]. Divergent demands for oxidative energy production and reductive metabolite formation may be accommodated by fission of mitochondria within a cell into a cristae- and ATP synthase-rich subpopulation versus a depleted subpopulation tuned for metabolite synthesis such as proline and ornithine production [109]. Disruption of mitochondrial fission fusion cycling could contribute imbalances such as the Warburg effect. A recent finding in twins with ME/CFS is that overexpression of Wiskott–Aldrich Syndrome Protein Family Member 3 (WASF3) can lead to endoplasmic reticulum stress and disrupt mitochondrial respiratory supercomplex formation and reduce complex IV levels [110]. Mitochondrial disruption has also been associated with age-associated pathologies, sarcopenia, insulin resistance and diabetes, anxiety, and affective disorders [111]. Specificity is an issue because it is not clear if there are distinct pathophysiological pathways in ME/CFS and each of the other disorders or if they all share a single consistent mechanism. A potential therapy is to use mitohormetic compounds to induce a mild reversible energetic stress. The increased demands deplete ATP and so lead to increased ADP/ATP ratios that activate the cellular energy sensor 5′-AMP-activated protein kinase (AMPK) [81]. AMPK protects cells from stresses that cause ATP depletion by switching off ATP-consuming biosynthetic pathways. This activates mitochondria quality control programs, leading to mitophagy and elimination of dysfunctional mitochondria. It is not yet clear if this therapy is appropriate for ME/CFS.

Other shifts were also apparent in ME/CFS. The elevation of trans-aconitate and citrate in the TCA cycle in ME/CFS is enigmatic. Trans-aconitate is reported to be an inhibitor of the Krebs cycle [112] and fumarase reaction between malate and fumarate [113]. Trans-aconitate can inhibit citrate dependent progestin synthesis in human placenta [114] and so may have other deleterious effects in vivo. Expression of aconitate hydratase is elevated in twins with ME/CFS [79]. Otherwise, this metabolite is reported to be a biomarker of soy ingestion from the diet. Trans-aconitate is the substrate of trans-aconitate 2-methyltransferase, which is found in fungi but not humans.

Can a shift in the gut microbiome and its metabolites cause fatigue and other brain manifestations in ME/CFS? Microbial metabolites were detected in cerebrospinal fluid. Microbial dysbiosis and reduced butyrate production are implicated in ME/CFS and irritable bowel syndrome [55,57]. The compounds include 2,3-butanediol, hippuric acid, flavone, indoles, phenylacetates, and other derivatives of phenylalanine, tyrosine, and tryptophan that are synthesized by gut microbes and processed in the liver [115]. Tryptophan-derived metabolites act through the aryl hydrocarbon receptor (AHR) to regulate gut dysbiosis and immunity [116]. Many are considered uremic toxins but have underappreciated effects in the brain and other organs [117]. Urea recycling was one of the enriched pathways, but ammonia was not measured [118]. N-Acetylglutamine was elevated and may act as a uremic toxin and neurotransmitter. Homocysteic acid was increased in ME/CFS and is a potent N-methyl-D-aspartate (NMDA) receptor agonist that is also found in astrocytes [119]. The concept of uremic toxicity, metabolism of essential aromatic amino acids, and aryl hydrocarbon receptors generates new possibilities for pathogenesis in ME/CFS.

Many of the metabolites were the end-products of their respective catabolic pathways. This raises the hypothesis that cells in ME/CFS secrete higher levels of catabolic end-products such as formate (serine), lactate (glycolysis, Warburg effect), trans-aconitate (TCA cycle), and nitrogenous compounds (creatine, creatinine, urea, ammonia, spermine, spermidine) compared to control. Production and excretion may be further elevated by cell stressors such as exertion.

Gender has a large impact on plasma [59] and cerebrospinal fluid constituents (Figure 4A). ME/CFS males had higher levels of metabolites than females with and without ME/CFS (Appendix A). Pathway enrichment in males selected pyrimidine metabolism, gluconeogenesis, Warburg effect, methionine metabolism, and spermidine and spermine biosynthesis. Males had more hexylceramides (HCER) than females (7 of 11 mass spectrometry targets, *p* = 0.0039 by Fisher exact test) while females had a trend for more phosphatidylglycerides (not significant) in the multivariate analysis.

The effects of exercise were compared between ME/CFS and SC in Figure 6 and Figure 8. ME/CFS and SC shared the consumption of phosphatidylglycerols, phosphatidylcholine, phosphatidylethanolamine, sphingomyelin, and triacylglycerides during exercise (non-exercise > post-exercise) (Appendix A).

In contrast, metabolite levels moved in opposite directions in ME/CFS and SC after exercise. At baseline (non-exercise), ME/CFS had buildups of metabolites suggesting blockages in consumption, excess synthesis in the brain, or importation into cerebrospinal fluid. Exercise caused the consumption of metabolites and significantly reduced levels in ME/CFS (ME/CFS non-exercise > post-exercise). This outcome was in contrast to previous findings of higher urine metabolites [68] and plasma lactate, acetate, methylhistidine, and methionine [67] in ME/CFS after exercise. In contrast, exercise induced higher levels of metabolites in SC. The SC metabolites that were higher in the post-exercise period were considered to have increased production in the brain, influx from plasma into the cerebrospinal fluid, or reduced utilization or metabolism in the healthy brain.

Lysine is an essential amino acid [120] that was elevated in ME/CFS prior to exercise. It is metabolized through aminoadipate to enter the TCA cycle. Lysine is the source for carnitine and L-acetylcarnitine, which were elevated in ME/CFS and are required for fatty acid transport across peroxisomal and mitochondrial membranes for oxidation. Peroxisomal dysfunction is implicated in ME/CFS [70].

Dopamine and 5-hydroxytryptophan derived from tryptophan are hormones and neurotransmitters that are important for affect, reward, and attention. Relative deficits induced by exertion could contribute to post-exertional malaise (PEM) and symptom exacerbations.

Other metabolites that were consumed after exercise in ME/CFS were 1-methyl-L-histidine, xanthylic acid, uridine, and urea. 2-Hydroxy-3-methylbutyric acid is a product of ketogenesis and the metabolism of valine, leucine, and isoleucine.

Post-exercise, there were fewer significant differences between ME/CFS and SC except for elevated levels in SC for phenylalanine, tyrosine, riboflavin, and 1-methylhistidine. Phenylalanine was altered in different directions with ME/CFS non-exercise > post-exercise but SC post-exercise > non-exercise.

Glutathione (reduced) rose after exercise, which may reflect oxidant generation during exertion.

Riboflavin was highest in SC post-exercise and was higher than ME/CFS. It is the precursor for flavin mononucleotide and flavin adenine dinucleotide (FAD) used for redox, TCA, and mitochondrial electron transport chain reactions [121].

Thiamine became elevated in cerebrospinal fluid following exercise in both ME/CFS and SC, suggesting its exit from brain cells or importation from plasma through the choroid plexus in all subjects. Elevated thiamine may be a marker of exercise. Thiamine monophosphate is the transported version of thiamine in plasma and cerebrospinal fluid, but it is not clear why it would be secreted from cells after exercise [122]. Thiamine is a cofactor for transketolase reactions, glycolysis, and the synthesis of creatine from glycine and methionine. Of note, creatinine was elevated before exercise in ME/CFS. Acute borderline intracellular deficiency of thiamine has been proposed to contribute to autonomic dysfunction, sleep apnea, and early signs suggestive of subclinical beriberi [122].

The metabolic alterations after exertion generates the hypothesis that simultaneous shifts in thiamine, folate, biotin, dopamine, 5-hydroxytryptophan, homocysteic acid, and other neurotransmitters and metabolites caused by exertion may have synergistic effects that destabilize neural pathways and contribute to central mechanisms of post-exertional malaise in ME/CFS.

Lactate was predicted to be elevated in ME/CFS based on molecular spectroscopy studies of ventricular fluid [123,124,125,126], elevated resting plasma levels [127], and potentially by diffusion into cerebrospinal fluid after exercise such as cardiopulmonary exercise stress testing which increases plasma lactate levels [128]. SC had an increase after exercise (Figure 8B) but there were no changes for ME/CFS. This may be because the submaximal exercise was insufficient to reach anaerobic threshold and excess lactate production. Alternatively, lactate metabolism in the brain is different from skeletal muscle because it is a fuel for neuron energy metabolism and so is rapidly sequestered as opposed to being excreted as in skeletal muscle [129,130,131]. The time course for normalization of cerebrospinal fluid lactate after exercise is not known, and so the recovery period may have been too long to detect transient alterations.

The findings in this study may be applicable to related disorders. Long COVID subjects have significant reduction in sarcosine and serine concentrations that were inversely correlated to depression, anxiety, and cognitive dysfunction scores [132]. Male Gulf War Illness veterans have significant elevation of ceramides, sphingomyelins, and phosphatidylcholines, which account for 78% of the metabolic impact [133]. GWI and ME/CFS males shared reductions in the purine pathway. Conversely, ceramide, sphingomyelin, phospholipid, and branched-chain amino acid pathways were increased in GWI but decreased in ME/CFS. This fits with other findings, such as activation of the dorsal midbrain during a cognitive provocation [103] that indicate comparable pathways are involved in the two diseases but with opposite directions of change. Fibromyalgia has similar patterns of change in gut microbiome [134]; higher plasma levels of serine, tyrosine, other amino acids, and urea; and lower levels of choline, glutamine, 2-hydroxyisocaproate, and pyruvate compared to controls [135,136]. The metabolic similarities in these diseases suggest common themes in disease pathogenesis that may lead to a better understanding of their underlying pathophysiologic mechanisms.

This study has a number of limitations. There was a relatively small sample size with unequal numbers per group that may have generated unbalanced outcomes. The non-exercise and post-exercise cohorts had different subjects. The protocols were not designed to perform lumbar puncture before and after exercise and so paired analysis in the same individuals was not possible. This raises the potential issue of systematic errors due to batch effects with differences in sampling or specimen storage. However, the same neuroradiology protocol, equipment, and standard operating procedures were used for both cohorts. All of the specimens were used without thawing. Additionally, the raw data were reconciled for batch effects in MetaboAnalyst. A subset of samples from the non-exercise group had previously been tested with a precursor of the current kit. Those results were equivalent to the current ones, indicating no deterioration of samples during storage [137]. Metabolites could increase after exercise because of increased passage from plasma across the choroid plexus, release from neurons and glia, or decreased uptake and intracellular metabolism. There is little information on the turnover of metabolites in these processes or the duration of the recovery periods after exercise. Better knowledge about the flux of the metabolites and lipids in cerebrospinal fluid will influence interpretation of the results. The small number of metabolite targets compared to shotgun or larger screening assays limited the number of pathways that could be matched and their levels of significance during enrichment. Therefore, *p*-values were not adjusted for multiple comparisons. Despite this limitation, the lists of lipids, amino acids, nucleotides, and other metabolites could be directly linked to the KEGG pathways and metabolomic map (Figure 2) and were consistent with the prior published knowledge. Multivariate analysis was corrected for multiple comparisons using the Sidak method.

Bayesian and multivariate analyses were purposefully performed in parallel to contrast the yield of significant metabolites and lipids found between ME/CFS and SC and the effects of exercise. The directions of significant differences between groups, effects of exercise and pathway enrichment analyses were consistent. The frequentist approach identified analytes that were significantly different between the two cohorts. The Bayesian approaches provide insights for significant relationships in the independent and linear regression analyses that can now be used as prior evidence for planning future studies. Incorporating the current data (Appendix A) with previous datasets may lead to stronger insights after meta-analysis and receiver operating characteristics for single compounds or biosignature panels.

## 4. Methods

### 4.1. Subjects

Protocols were approved by the Georgetown University Institutional Review Board (2006-481, 2009-229, 2013-0943, 2015-0579) in accordance with the Declaration of Helsinki and listed in clinicaltrials.gov (NCT03567811, NCT00810329).

Sedentary ME/CFS and healthy control subjects were recruited using websites, word of mouth, fliers, newspaper and online advertisements, and personal contacts in clinics and support groups. Interested participants responded via telephone or email. After obtaining verbal informed consent, each volunteer had an initial telephone screening with a clinical research associate, who read a scripted outline of the study to assess inclusion and exclusion criteria. Candidates were screened for Centers for Disease Control (CDC) criteria for Chronic Fatigue Syndrome (CFS) [1]; current medications; chronic medical and psychiatric illnesses [138,139,140]. Eligible subjects came to the Georgetown Howard Universities Clinical and Translational Science Clinical Research Unit, where diagnosis and study inclusion were confirmed by history and physical examination after subjects provided in-person written informed consent.

ME/CFS was defined using 1994 CDC “Fukuda” criteria [1] plus Canadian Consensus Criteria [2,3]. The CDC criteria require disabling fatigue lasting more than 6 months that cannot be explained by exclusionary medical or psychiatric diagnoses [8,9,10,141] plus at least 4 of 8 ancillary symptoms: short-term memory or concentration problems, sore throat, sore lymph nodes, myalgia, arthralgia, headache, sleep disturbance, and post-exertional malaise (exertional exhaustion). The Carruthers Canadian Consensus Criteria emphasizes fatigue, post-exertional malaise, sleep, pain, and cognition, and an added array of flu-like, autonomic, and interoceptive symptoms [2,3]. Subjects were not retrospectively assessed for 2015 Institute of Medicine criteria for Systemic Exertion Intolerance Disease (SEID) with fatigue, post-exertional malaise, sleep, and either cognitive or orthostatic problems [4].

Sedentary controls (SC) in the non-exercise and post-exercise groups had a sedentary lifestyle with less than 40 min of aerobic exercise per week and did not meet ME/CFS criteria. They met the exclusion criteria and so were sedentary but otherwise healthy. Controls may have had some mild complaints or individual symptoms such as isolated moderate or severe fatigue (e.g., chronic idiopathic fatigue) but did not meet the Fukuda or Canadian ME/CFS criteria.

Exclusion criteria included substance abuse, hospitalization for a psychiatric disorder in the past 5 years, or a chronic medical or psychiatric condition [138,139,140].

Two cohorts were studied [14,15,16,17]. The first day of the 2 protocols was considered an adjustment period, and included the patient’s history and physical, blood work, and baseline studies. The non-exercise cohort rested before having lumbar puncture and did not have exercise. The post-exercise cohort had magnetic resonance imaging (MRI), submaximal bicycle exercise stress testing, and serial assessments of postural tachycardia. They rested overnight, then had their second identical stress test, MRI, and post-exercise lumbar puncture. Subjects cycled at 70% of predicted heart rate (pHR = 220-Age) for 25 min then increased to 85% pHR. Lumbar puncture was performed 1 to 5 h after exercise.

### 4.2. Questionnaires

Subjects completed a series of questionnaires (Appendix A). Disability and impairment were assessed based on quality of life and the Medical Outcomes Survey Short Form 36 (SF-36) [142,143]. Responses were converted from nominal and anchored ordinal scores to scales from 0 (severely impaired) to 100 (no impairment) [144]. The average of Vitality, Role Physical, and Social Functioning (SF36 V,RP,SF) was calculated as it consistently gave the lowest scores and was superior to individual domains for differentiating ME/CFS and GWI from SC.

All subjects completed the CFS Symptom Severity Questionnaire (CFSQ) [72] and were operationalized by scoring fatigue and 8 ancillary symptoms from the previous 6 months on an anchored ordinal scale with grades of none = 0, trivial = 1, mild = 2, moderate = 3, and severe = 4. Unlike the original Fukuda criteria, we required moderate or severe symptom severities for fatigue and at least four of the eight criteria in order to be considered for ME/CFS diagnosis here.

Fatigue was corroborated using independent scales. The Revised Clinical Interview Schedule (CIS-R) [145] consists of 6 topics used for interviews and was adapted as 6 nominal items to gauge overall fatigue and tiredness (range 0 to 6). The Chalder Fatigue questionnaire was assessed as the total score summed for 11 items (range 0 to 33) [146]. The Multidimensional Fatigue Inventory (MFI) addressed five domains with ranges of 0 to 20 and sum (0 to 100) [147,148].

The McGill pain questionnaire [148] scored 11 “Sensory” pain descriptors and four “Affective” words (tiring, sickening, fearful, punishing). Severity was graded on an anchored ordinal scale, None = 0, Mild = 1, Moderate = 2, and Severe = 3, and summed for “Sensory” (range 0 to 33), “Affective” (range 0 to 12), and “Total” (range 0 to 45) scores.

Interoceptive symptoms were assessed from several questionnaires. Chronic Multisymptom Severity Inventory (CMSI) assessed interoceptive symptoms using the 0- to 4-point anchored ordinal scale [149]. Domain scores were determined for rheumatic symptoms, which included pain and fatigue symptoms (range 0 to 44), dyspnea (range 0 to 20), cardiac (range 0 to 16), headache (migraine and tension scored 0 to 4 each), ear sinus (range 0 to 20), neuro (range 0 to 16), irritable bowel syndrome based on Rome I criteria (range 0 to 32) [88], bladder (range 0 to 16), and the sum of all items (range 0 to 172). An interoceptive CMSI score (CMSI no pain) was calculated by subtracting the rheum domain from the sum of the CMSI (range 0 to 128). Migraines were assessed by International Headache Society criteria [150]. Upper and lower airway symptoms were assessed using the Rhinitis Score [149] and Irritant Rhinitis Score [151]. Systemic irritant symptoms were assessed using the Chemical Exposures questionnaire domain scores [152]. The Composite Autonomic Symptom Score (COMPASS-31) graded symptoms conveyed by cranial nerve, autonomic, and general afferent system pathways which are commonly associated with autonomic dysfunction [153].

Psychiatric disorders were screened by PRIMEMD questionnaire for major depressive syndrome, other depressive syndrome, panic syndrome, and difficulty in completing the questionnaire [154]. Details of anxiety were examined with the Generalized Anxiety Disorder 7 questionnaire (GAD7) [155], Mood and Anxiety Questionnaire (MASQ) [156], and the Irritability Questionnaire [157]. Major depression [158] and somatic, anhedonia, and depressed domains were probed with the Center for Epidemiology—Depression questionnaire [159,160]. Scores ≥ 16 out of 60 have been used to infer risk of major depression but this is biased by somatic complaints such as fatigue that are common in major depressive disorder in the general population [161] but are also inherent to CFS criteria and diagnosis.

Psychological aspects of pain were examined with the Pain Beliefs and Perceptions questionnaire [162], Beliefs in Pain Control Questionnaire (BPCQ) [163], Pain Catastrophizing Scale [164], and “Your Experiences with Pain” (Chronic Pain Stressor Scale, CPSS) [165]. Posttraumatic stress disorder (PTSD) was assessed by PTSD Check List Civilian (PCL-C) [166].

### 4.3. Dolorimetry

Central sensitization was assessed by dolorimetry. Dolorimetry was performed with a strain gauge (DPP gauge; Chatillion Products, Ametek Inc., Largo, FL, USA) fitted with a 1 cm^2^ rubber stopper with pressure applied at a rate of 0.5 to 1 kg/s against the 18 traditional tender points [167,168,169]. The outcome point was the pressure that caused the subject to state that he/she was experiencing pain. A key aspect was to ensure that the patient felt in control of the process and had trust that the operator would stop pressing as soon as she indicated pain had developed. The mean of the 18 measurements was the dolorimetry pressure threshold. The coefficient of variability for dolorimetry was 9.3% for 57 women and 12.5% for 58 men who had serial measurements on 3 days by different staff members [169]. The Pearson correlation coefficient between thumb pressure tender point counts and dolorimetry pressure thresholds was −0.862 (explained variance = 0.742).

### 4.4. Lumbar Puncture

Identical methods were used in all subjects. Cerebrospinal fluid (20 mL) was drawn from the L1-L2 interspace using Gertie Marx needles (IMD Inc. Katonah, NY, USA) [170,171] in the prone position under fluoroscopic guidance by interventional radiologists. Specimens were immediately placed on ice, centrifuged at 4 °C, sent for routine laboratory studies, aliquoted, and stored at −80 °C within 1 h. After lumbar puncture subjects were strictly instructed to rest in comfortable positions and to avoid straining actions and Valsalva maneuvers such as lifting luggage for 24 h before discharge.

### 4.5. Metabolomics

Targeted mass spectrometry approaches developed at Georgetown University [172,173,174,175,176] were used to assess 179 metabolites and 220 lipids in cerebrospinal fluid in the non-exercise and post-exercise cohorts of ME/CFS and SC subjects.

All LC-MS-grade solvents, including acetonitrile and water, were purchased from Fisher Optima grade, Fisher Scientific (Pittsburgh, PA, USA). High-purity formic acid (99%) was purchased from Thermo-Scientific (Waltham, MA, USA). Debrisoquine and 4-nitrobenzoic acid were purchased from Sigma-Aldrich (Burlington, MA, USA). EquiSPLASH^®^ LIPIDOMIX^®^ quantitative mass spec internal standard and 15:0-18:1-d7-PA, C15 Ceramide-d7 (d18:1-d7/15:0), and 18:1 Chol (D7) ester were purchased from Avanti polar lipids. Internal standards for free fatty acid (FFA), dihydroceramides (DCERs), hexosylceramides (HCERs), and lactosylceramides (LCERs) were purchased from Sciex (Framingham, MA, USA) as the Lipidyzer platform kit.

### 4.6. Metabolomics and Lipidomics Protocols

The targeted metabolomics method quantitated 270 endogenous metabolites using a QTRAP^®^ 5500 LC-MS/MS System (Sciex, Framingham, MA, USA). Cerebrospinal fluid samples were thawed; then, 150 μL was immediately mixed with 150 μL of methanol/water 50/50, thoroughly mixed on a vortex mixer, incubated in an ultrasonic bath for 5 min, and incubated on ice for 20 min. A 150 μL quantity of acetonitrile was added, followed by incubation at −20 °C for 20 min. After vortexing, samples were centrifuged at 13,000 rpm for 20 min at 4 °C. The supernatant was evaporated to dryness under nitrogen at 30 °C using a vacuum pump. The residue was reconstituted to 150 μL by adding 150 μL of 25 MeOH/25 water/50 acetonitrile containing 250 ng/mL of debrisoquine (DBQ) as internal standard for positive mode and 250 ng/mL of 4-nitrobenzoic acid as internal standard for negative mode. The tubes were vortexed and incubated in the ultrasonic bath for 10 min then centrifuged at 13,000 rpm for 20 min at 4 °C. The supernatants were transferred to MS vials for LC-MS analysis.

The quality control sample was made with 25 µL National Institute of Standards and Technology standard plasma sample dissolved in 100 μL of extraction buffer (methanol/water 50/50) containing 250 ng/mL of debrisoquine (DBQ) as internal standard for positive mode and 250 ng/mL of 4-nitrobenzoic acid as internal standard for negative mode. The sample was vortexed for 30 s and incubated on ice for 20 min, followed by addition of 100 μL of acetonitrile and incubation at −20 °C for 20 min. Samples were centrifuged at 13,000 rpm for 20 min at 4 °C. The supernatant was transferred to MS vial for LC-MS analysis.

Five microliters of the prepared sample was injected onto a Kinetex 2.6 μm 100 Å 100 × 2.1 mm (Phenomenex, Torrance, CA) using an SIL-30 AC auto sampler (Shimazdu, Columbia, MD, USA) connected with a high-flow LC-30AD solvent delivery unit (Shimazdu) and CBM-20A communication bus module (Shimazdu) online with the QTRAP 5500 (Sciex) operating in positive and negative ion modes. A binary solvent comprising water with 0.1% formic acid (solvent A) and acetonitrile with 0.1% formic acid (solvent B) was used. The extracted metabolites were resolved at 0.2 mL/min flow rate. The liquid chromatography gradient conditions were as follows: initial—100% A, 0% B for 2.1 min; 14 min—5% A, 95% B until 15 min; 15.1 min—100% A, 0% B until 20 min. The auto sampler and oven were kept at 15 °C and 30 °C, respectively. Source and gas settings for the method were as follows: curtain gas  =  40; CAD gas  =  9; ion spray voltage  =  1700 V in positive mode and ion spray voltage  =  1600 V in negative mode; temperature  =  350 °C; ion source gas 1  =  30 and ion source gas 2  =  50. The data were normalized to internal standard area and processed using Sciex OS software 3.4. To ensure high quality and reproducibility of LC-MS data, the column was conditioned using the quality control samples initially and after every 10 sample injections to monitor shifts in signal intensities and retention time as measures of reproducibility and data quality. The NIST (National Institute of Standards and Technology, Gaithersburg, MD, USA) plasma sampling was run after every 20 samples to check the instrumental variance. Blank solvent was run after every 10 samples and before and after pooled quality control samples to minimize carry-over effects.

The targeted lipidomics method was designed to measure 21 classes of lipid molecules, including free fatty acids (FFA), diacylglycerols (DAG), triacylglycerols (TAG), phosphatidylglycerol (PG), phosphatidylcholine (PC), phosphatidylethanolamine (PE), phosphatidylinositol (PI), phosphatidylserine (PS), phosphatidic acid (PA), lysophosphatidylcholine (LPC), lysophosphatidylethanolamine (LPE), lysophosphatidylinositol (LPI), lysophosphatidic acid (LPA), ceramides (CER), dihydroceramides (DCER), hexosylceramide (HCER), lactosylceramide (LCER), sphingomyelins (SM), acyl carnitines, and cholesterol esters (CE) using the QTRAP^®^ 5500 LC-MS/MS System (Sciex).

Cerebrospinal fluid was thawed at room temperature and immediately processed by mixing 150 μL cerebrospinal fluid with 300 μL of isopropanol, vortexing serial incubations of 5 min in an ultrasonic bath, 20 min on ice, and incubation at −20 °C for 20 min. After vortexing, the samples were centrifuged at 13,000 RPM for 20 min at 4 °C. The supernatant was evaporated to dryness under nitrogen at 30 °C using a vacuum pump. Then, 150 μL isopropanol with internal standards was added to the residue to reconstitute the initial volume of CSF. The tubes were vortexed and incubated in the ultrasonic bath for 10 min. Samples were centrifuged at 13,000 rpm for 2 h at 4 °C. The supernatants were transferred to MS vials for LC-MS analysis.

NIST plasma sample (25 µL) was mixed with 125 μL of chilled isopropanol containing internal standards for lipid classes. The sample was vortexed for 30 s and incubated on ice for 30 min followed by incubation at −20 °C for 2 h. Samples were centrifuged at 13,000 rpm for 20 min at 4 °C. The supernatants were transferred to MS vials for LC-MS analysis.

Five microliters of each sample was injected onto an Xbridge amide 3.5 µm, 4.6 × 100 mm column (Waters, Milford, MA, USA) using an SIL-30 AC auto sampler (Shimazdu) connected with a high-flow LC-30AD solvent delivery unit (Shimazdu) and CBM-20A communication bus module (Shimazdu) online with the QTRAP 5500 (Sciex, MA, USA) operating in positive and negative ion modes. A binary solvent comprising acetonitrile/water 95/5 with 10 mM ammonium acetate as solvent A and acetonitrile/water 50/50 with 10 mM ammonium acetate as solvent B was used for the resolution. Lipids were resolved at 0.7 mL/min flow rate; initial gradient conditions started with 100% of solvent A, shifting towards 99.9% of solvent A over a time period of 3 min, 94% of solvent A over a time period of 3 min, and 25% of solvent A over a period of 4 min. Finally, the column was washed with 100% of B for 6 min and equilibrating to initial conditions (100% of solvent A) over a time period of 6 min using auto sampler temperature 15 °C and oven temperature 35 °C. Source and gas settings were as follows: curtain gas  =  30; CAD gas  =  medium; ion spray voltage  =  5.5 kV in positive mode and −4.5 kV in negative mode; temperature  =  550 °C; nebulizing gas  =  50; and heater gas  =  60. The data were normalized to respective internal standard area for each class of lipid and processed using MultiQuant 3.0.3 (Sciex). The quality and reproducibility of LC-MS data were ensured using a number of measures. The column was conditioned using the pooled QC samples initially and after every 10 sample injections. The NIST plasma sample was run after every 20 samples to check the instrumental variance. Solvent blanks were run after every 10 samples and before and after pooled quality control samples to minimize carry-over effects.

The primary data are provided in Appendix A.

### 4.7. Statistical and Bioinformatic Analysis

Batch effects were examined using the COMBAT program in MetaboAnalyst 6.0 [177,178] and the reconciled non-exercise and post-exercise datasets were used. Zeros were found in about 20 cells and one half the minimum value for each analyte was imputed. Data were log 10 transformed and autocorrected by subtracting the mean and dividing by the standard deviation. Data and hypotheses were assessed by a series of frequentist and Bayesian methods to compare their outcomes [18].

Data were assessed in SPSS version 29 (IBM, Armonk, NY, USA) [179] by multivariate general linear modeling with metabolites and lipids as dependent variables; disease (ME/CFS vs. SC), gender and exercise status (non-exercise vs. post-exercise) as independent fixed factors; and age [59,180,181,182] and body mass index (BMI) [54,55,56] as covariates, and corrected by the Sidak method (*p* < 0.05) to control for multiple comparisons. Estimated marginal means were evaluated for each factor and their triple product (disease × gender × exercise). Significant differences between ME/CFS vs. SC for the main effect of disease and cross-products were tabulated.

Relationships were examined by pathway analysis and enrichment in MetaboAnalyst 6.0. Many of the secondary and less well-studied metabolites were not linked to KEGG pathways and so their functions were detailed from the Human Metabolome Database MetaboCards [88,183]. Previous studies in ME/CFS have identified alterations in metabolic modules for glycolysis, tricarboxylic acid cycle (TCA), anaplerotic and other amino acid interactions with the TCA cycle, purine and pyrimidine metabolism, and dysfunctional peroxisome and mitochondrial activities. In addition, our multivariate findings identified serine, 5-methyltetrahydrofolate (5MTHF), and other derivatives of the serine–glycine–one-carbon pathway. The serine pathway provided a link to rationalize phospholipid synthesis in the setting of ME/CFS. KEGG pathways were used to integrate these metabolic modules into an interconnected general framework (Figure 2). Metabolic modules were color-coded. Metabolites that were significantly different were indicated in bold ± italics and highlighted for ME/CFS > SC (yellow) and SC > ME/CFS (pink). Metabolites derived from the gut microbiome, diet, exposome, or with poorly defined metabolism in humans were tabulated.

### 4.8. Bayesian Analysis

Bayesian linear regression with disease, gender, exercise, age, and BMI as covariates was performed as a comparison to the multivariate frequentist approach. Default prior evidence was used in SPSS, as there were no previous cerebrospinal fluid studies using the current methods and panels. Significant differences in posterior probabilities for each independent variable were defined by posterior modes and 95% credible intervals that excluded zero.

The same log autocentered data were analyzed by four separate Bayesian analyses for independent samples with age and BMI as covariates in order to show differences at baseline (non-exercise ME/CFS vs. SC), post-exercise (post-exercise ME/CFS vs. SC), and the effects of exercise on ME/CFS (non-exercise vs. post-exercise) and SC (non-exercise vs. post-exercise). Significant outcomes for differences by disease and exercise status were mapped onto the multivariate metabolomic framework of Figure 2 to facilitate visual comparisons of metabolites that were up- or downregulated. Results were analyzed by MetaboAnalyst pathway and enrichment analysis, which expanded the scope of the metabolomic differences and implicated pathways.

These four Bayesian comparisons were mimicked by performing frequentist multivariate analyses with disease as the independent factor, age and BMI as covariates, but without gender in order to compare the outcomes of the two statistical philosophies. Outcomes from each method were tabulated (Appendix A) and sorted to identify the intersecting sets of metabolites and lipids and their relationships to each of the independent variables. This allowed the comparison of the significantly different point estimates between ME/CFS and SC from the multivariate results, and the metabolites and lipids that best supported the hypothetical separation between ME/CFS and SC using Bayesian results. Three separate sorting steps focused on disease (ME/CFS vs. SC), gender (male vs. female), and non-exercise vs. post-exercise differences.

## 5. Conclusions

Logarithmic transformation, normalization, and standardization of raw mass spectrometry abundances for multivariate general linear modeling increased the yield of significantly elevated metabolites and lipids in ME/CFS cerebrospinal fluid and the differences related to exercise, gender, age, and BMI. The elevation of serine in cerebrospinal fluid in ME/CFS was a key finding because of the many metabolic relationships that were uncovered. Elevated serine was associated with decreased 5MTHF, indicating disruption of one-carbon metabolism in the brain. This was supported by elevation in sarcosine, creatine, purines and thymidine derivatives. Energy metabolism was disrupted as indicated by elevated trans-aconitate. Serine is the precursor of phospholipids which were also increased in ME/CFS. Increased sphingomyelins and hexylceramides implicated brain white matter dysfunction and supported alterations found in fMRI studies [23,25,104,105] and the prolonged response times during the cognitively challenging Stroop task [184]. Exercise led to consumption of lipids in ME/CFS and SC, but metabolites were consumed by ME/CFS and produced in SC. The “hypermetabolic” state is at variance with other studies of cerebrospinal fluid in ME/CFS [31,32], but the general categories of metabolic dysfunction align with the marked dysfunction of ME/CFS.

## Figures and Tables

**Figure 1 ijms-26-01282-f001:**
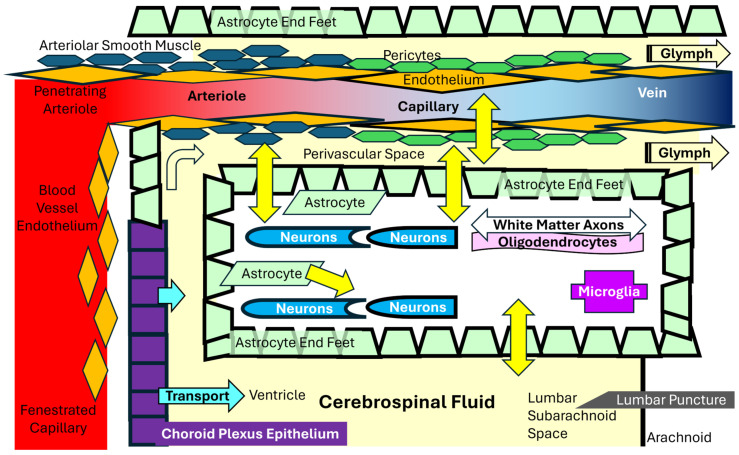
Generation of cerebrospinal fluid. Plasma flux through the blood–brain parenchyma and blood–choroid plexus barriers, brain metabolism, and the exchange of solutes between brain interstitial fluid and cerebrospinal fluid determine the composition of cerebrospinal fluid. It is presumed that metabolic effects in brain cells can be inferred from the constituents in cerebrospinal fluid. Cerebrospinal fluid (light yellow) is synthesized by tightly controlled transport processes via choroid plexus epithelial cells. Cerebrospinal fluid fills the ventricular system and perivascular spaces. Plasma is separated from brain parenchyma by endothelial cells (orange), pericytes (green), arteriolar smooth muscle (dark blue), cerebrospinal fluid (light yellow) in the perivascular space, and astrocytic end feet processes (light green). Metabolites and lipids diffuse between astrocytes, neurons (light blue), microglia (magenta), oligodendrocytes (pink), and myelin in white matter (white). Brain interstitial fluid (black) exchanges metabolites between cells and fluid spaces and is reabsorbed into the venous system via glymphatic channels while cerebrospinal fluid is reabsorbed by arachnoid granulations. There is no guarantee that lumbar puncture can sample all of these compartments and nutrient exchanges.

**Figure 2 ijms-26-01282-f002:**
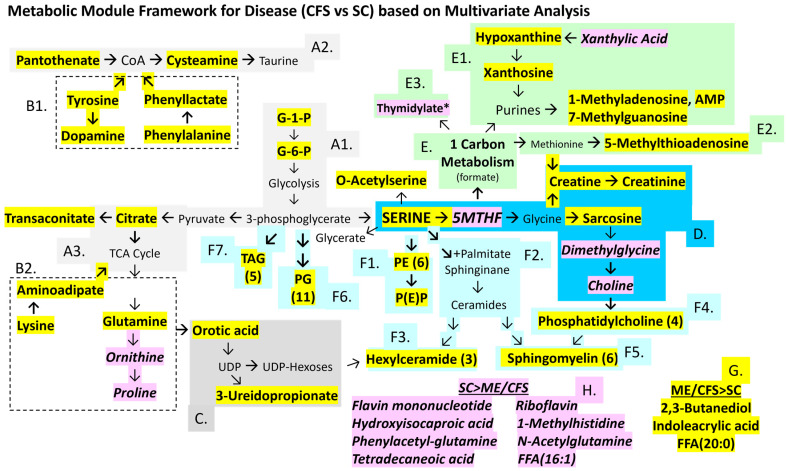
Metabolic modules and imbalances in non-exercise ME/CFS. The map provides an overview of cerebrospinal fluid metabolic pathways that differed between ME/CFS and SC at baseline and were influenced by exercise in multivariate analysis. Metabolic pathways are summarized (indicated by arrows connecting strings of intermediate compounds) and organized into color-coded modules. Metabolites that were significantly elevated in ME/CFS (ME/CFS > SC) are in bold with yellow highlighting while the converse (SC > ME/CFS) are in italics with pink highlighting. Central energy metabolism is shown in grey with A1. glycolysis, A2. coenzyme A (CoA) metabolism, and A3. TCA (tricarboxylic acid) cycle. Amino acid metabolism (beige) has two components: B1. essential aromatic amino acid metabolism that leads to acetylCoA and B2. (boxes with dashed lines) anaplerotic amino acid metabolism related to the TCA cycle. Glutamine plays a central role in these pathways and C. pyrimidine metabolism (dark grey). A novel finding was the persistent finding of serine (bold, upper case, yellow highlighting) in ME/CFS and the number of metabolites in the D. serine/glycine/5-methyltetrahydrofolate (5MTHF)/one-carbon pathways (dark blue). E. Methyl transfer (green) is essential for E1. purine synthesis, E2. methionine pathway, and E3. thymidylate synthesis. Serine provides a direct link to phospholipid synthesis (F., blue) as a source material for phosphatidylethanolamine (PE) and phosphatidylethanolamine plasmalogens (P(E)P) (F1). F2. Serine and palmitate combine to form sphinginane, which is the precursor of ceramides. F3. UDP-hexoses are attached to form hexylceramides. F4. Glycine is metabolized by N-methylation to form sarcosine, dimethylglycine, and choline (trimethylglycine) for synthesis of phosphatidylcholines (PCs). F5. Phosphatidylcholines exchange headgroups with ceramides to form sphingomyelins (SMs). F6. 3-Phosphoglycerate and dihydroxyacetone phosphate are source materials for the formation of phosphatidylglycerols (PGs) and F7. triacylglycercides (TAGs). Numbers in parentheses after lipids indicate the number that were significantly altered (*p* < 0.05, Sidak correction for multiple comparisons). Metabolites with other physiologies or microbial origins are tabulated for ME/CFS > SC (G., yellow) and SC > ME/CFS (H., pink). Other abbreviations: G-1-P: glucose-1-phosphate; G-6-P: glucose-6-phosphate; AMP: adenosine monophosphate. Nonsignificant intermediates are in smaller font (not bold). * Thymidylate is a special case because different derivatives from the mass spectrometry analysis were altered in different directions.

**Figure 3 ijms-26-01282-f003:**
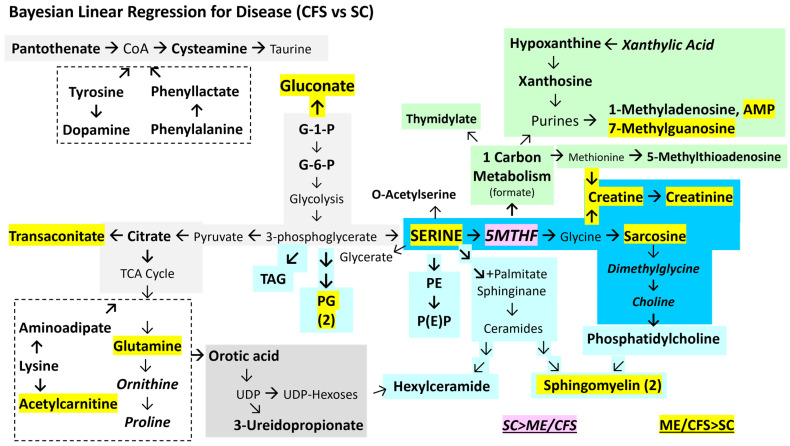
Bayesian linear regression. Serine, sarcosine, creatine, creatinine, trans-aconitate, and other metabolites were elevated in ME/CFS cerebrospinal fluid. 5MTHF was the only metabolite to be elevated in SC > ME/CFS. Nine metabolites (22%) were shared with the frequentist result multivariate linear regression. Annotation as per Figure 2.

**Figure 4 ijms-26-01282-f004:**
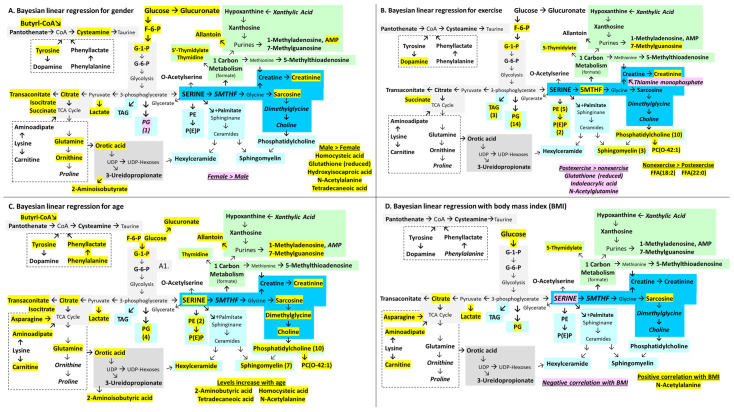
Bayesian linear regression factors. Metabolites and lipids that were significant in Bayesian linear regression were compared to the metabolic map (Figure 2) for (**A**) gender, (**B**) exercise, (**C**) age, and (**D**) body mass index (BMI). Annotation as per Figure 2.

**Figure 5 ijms-26-01282-f005:**
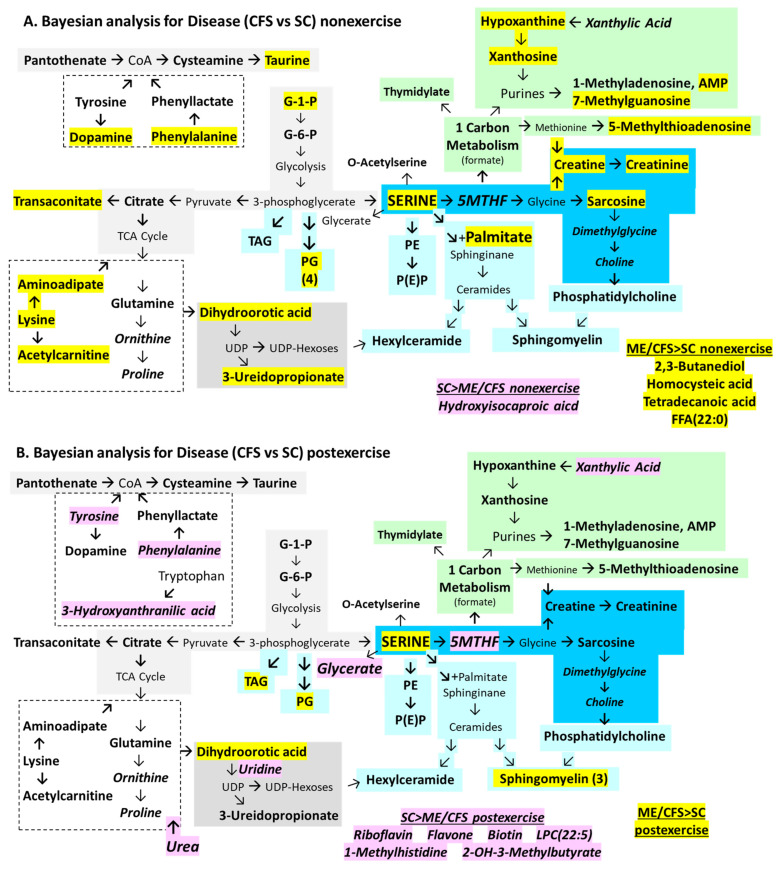
Bayesian analysis of ME/CFS vs. SC in non-exercise and post-exercise cohorts. (**A**) Metabolites that were significantly elevated before exercise in ME/CFS (ME/CFS > SC non-exercise) are shown in bold and highlighted in yellow. Only hydroxyisocaproic acid was higher in SC > ME/CFS (pink). (**B**) The post-exercise comparison of ME/CFS vs. SC found elevated serine in ME/CFS (yellow highlighting) with reciprocal lower 5MTHF and other vitamins (SC > ME/CFS post-exercise, pink italics). Modules are color-coded as in Figure 2.

**Figure 6 ijms-26-01282-f006:**
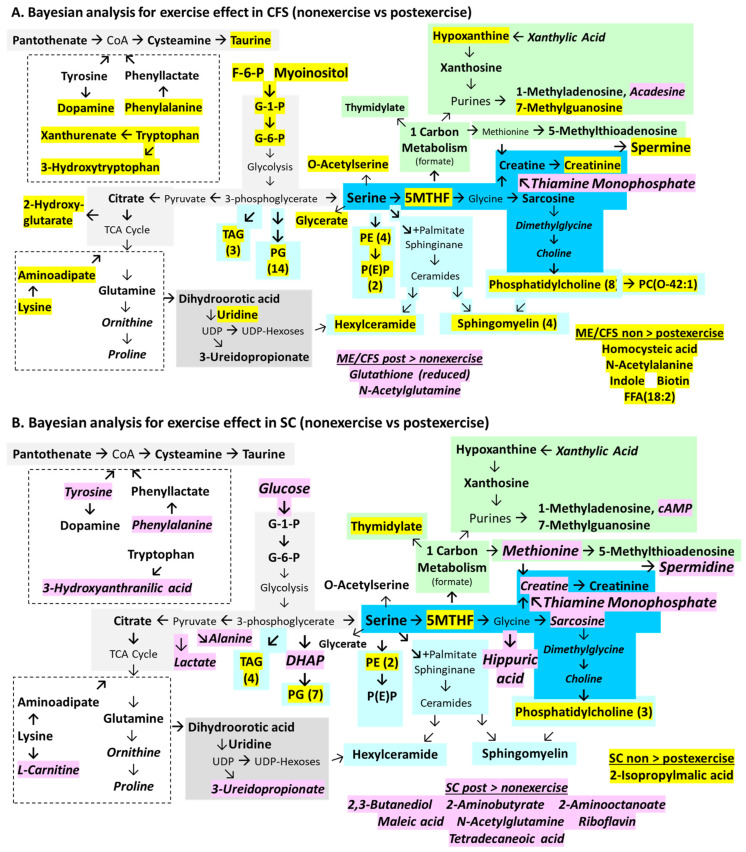
Bayesian analysis of the effects of exercise on ME/CFS and SC. (**A**) Exercise had significant effects on ME/CFS in the non-exercise > post-exercise (yellow) and post-exercise > non-exercise (pink) conditions. (**B**) SC had higher 5MTHF and lipids in non-exercise samples. However, more metabolites were elevated after exercise (SC post-exercise > non-exercise, pink). Annotation as per Figure 2.

**Figure 7 ijms-26-01282-f007:**
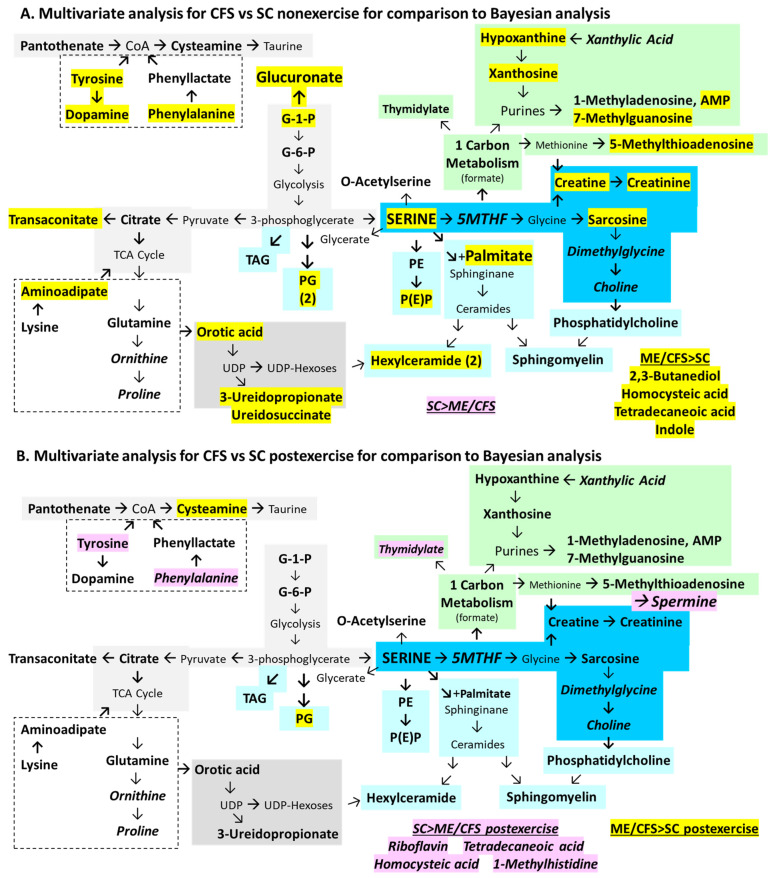
Multivariate analysis for ME/CFS vs. SC in the non-exercise and post-exercise data. This frequentist analysis did not include gender so it could be compared to Bayesian outcomes (Figure 5). (**A**) ME/CFS had higher concentrations than SC in the non-exercise condition. (**B**) Only eight metabolites were significantly different between ME/CFS and SC in the post-exercise cohort, but four of them matched Figure 5B. Annotation as in Figure 2.

**Figure 8 ijms-26-01282-f008:**
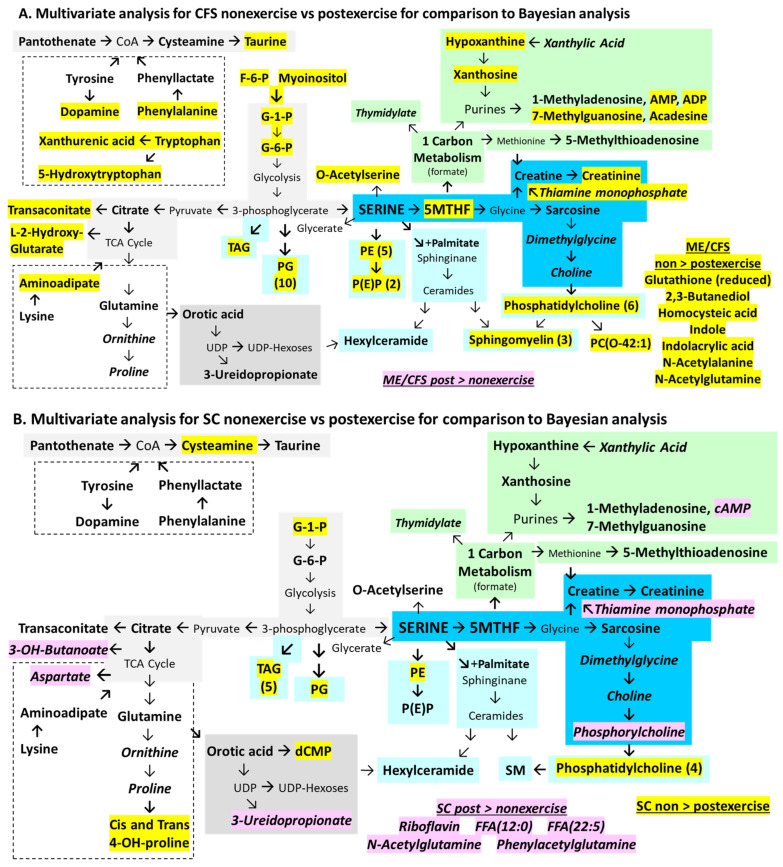
Frequentist multivariate analysis of non-exercise vs. post-exercise for ME/CFS and SC for comparison to Figure 6. (**A**) ME/CFS had 30 metabolites that were consumed during exertion. (**B**) SC had six metabolites and eleven lipids that were consumed (yellow) and nine metabolites that accumulated or were produced in excess after exercise (pink). Annotation as per Figure 2.

**Table 1 ijms-26-01282-t001:** CONSORT information.

Study Protocol	Non-Exercise Protocol with Lumbar Puncture	Exercise Followed by Lumbar Puncture
Contacted	271	148
Screened	100	73
Medical exclusions	13	11
Declined to participate	22	35
Lumbar puncture	65 (45 female)	27 (11 female)
Groups and N (N females)		
Sedentary control (SC)	20 (9 female)	12 (2 female)
ME/CFS	45 (36 female)	15 (9 female)
Total	92 (56 female)	

**Table 2 ijms-26-01282-t002:** Demographics. Mean ± SD.

Mean SD	Control	Control	ME/CFS	ME/CFS
	Non-exercise	Post-exercise	Non-exercise	Post-exercise
N	20	12	45	15
Female	9	2	36	9
Age	43.3 ± 12.5	46.7 ± 9.9	45.7 ± 11.0	45.0 ± 10.2
BMI	27.2 ± 5.5	30.5 ± 4.8	28.7 ± 7.2	27.3 ± 5.8
FM 1990	15%	8%	44%	13%
FM 2010	5%	0%	50%	60%
CFS Questionnaire (range 0 to 4). All scores for ME/CFS were significantly higher than SC (*p* < 0.001).				
Fatigue	1.3 ± 1.3	1.2 ± 1.0	3.7 ± 0.5	3.7 ± 0.6
PEM	1.1 ± 1.6	0.4 ± 0.8	3.4 ± 0.9	3.5 ± 0.5
Cognition	1.0 ± 1.2	1.0 ± 1.3	3.1 ± 0.8	2.8 ± 0.7
Sleep	0.7 ± 1.1	1.1 ± 1.2	2.5 ± 1.2	1.6 ± 1.3
Myalgia	1.1 ± 1.4	0.9 ± 1.0	2.9 ± 1.3	2.5 ± 1.3
Arthralgia	0.7 ± 1.0	1.1 ± 1.0	2.5 ± 1.2	2.2 ± 1.4
Headache	1.3 ± 1.5	1.8 ± 1.3	3.5 ± 0.9	3.4 ± 0.6
Sore throat	0.4 ± 0.7	0.3 ± 0.6	1.5 ± 1.2	1.6 ± 1.0
Sore nodes	0.3 ± 0.6	0.1 ± 0.3	1.3 ± 1.2	1.1 ± 1.1

**Table 3 ijms-26-01282-t003:** Cerebrospinal fluid chemistries. There were no erythrocytes or leukocytosis.

	SC Mean ± SD	ME/CFS Mean ± SD	
N	32	60	
Total protein	35.8 ± 11.5	34.4 ± 11.1	16 to 58 mg/dL
Albumin	23.3 ± 8.6	20.2 ± 7.9	8 to 37 mg/dL
Alb CSF/serum	6.2 ± 2.1	5.1 ± 1.9	<9
IgG	2.6 ± 1.1	2.4 ± 1.0	1 to 4 mg/dL
Glucose	63.0 ± 8.4	61.9 ± 8.8	50 to 75 mg/dL

**Table 4 ijms-26-01282-t004:** Student’s *t*-tests (uncorrected) for non-transformed mass spectrometry abundances of metabolites and lipids between non-exercise SC and ME/CFS.

Analyte	SC Mean ± SD *n* = 20	ME/CFS Mean ± SD *n* = 45	P	Hedges’ g
ME/CFS > SC				
Serine	0.0134 ± 0.0023	0.0156 ± 0.0023	0.00094	0.933
7-Methylguanosine	0.0492 ± 0.0093	0.0573 ± 0.0112	0.0062	0.762
Ureidopropionic acid	0.0170 ± 0.0027	0.0199 ± 0.0041	0.0066	0.756
Aminoadipate	0.0043 ± 0.0008	0.0052 ± 0.0014	0.0081	0.736
Homocysteic acid	0.0116 ± 0.0023	0.0135 ± 0.0028	0.014	0.680
Creatinine	1.9051 ± 0.2449	2.0979 ± 0.3123	0.017	0.658
Creatine	0.9578 ± 0.1904	1.0941 ± 0.2262	0.022	0.631
1-Methyladenosine	0.0121 ± 0.0052	0.0156 ± 0.0061	0.031	0.594
Palmitic acid	0.3193 ± 0.0323	0.3372 ± 0.0294	0.031	0.591
Xanthosine	0.0088 ± 0.0025	0.0102 ± 0.0026	0.034	0.582
Taurine	0.0586 ± 0.0123	0.0676 ± 0.0175	0.041	0.563
Trans-Aconitate	0.2807 ± 0.0396	0.3151 ± 0.0689	0.041	0.561
Dopamine	0.0229 ± 0.0047	0.0257 ± 0.0053	0.043	0.554
Methylthioadenosine	0.1555 ± 0.0570	0.1860 ± 0.0544	0.044	0.552
2,3-Butanediol	0.0997 ± 0.0121	0.1116 ± 0.0248	0.047	0.547
Tetradecanedioic acid	0.0113 ± 0.0013	0.0120 ± 0.0015	0.049	0.539
SC > ME/CFS				
Hydroxyisocaproic acid	0.5344 ± 0.1621	0.4438 ± 0.1073	0.0097	0.715
L-Ornithine	0.0045 ± 0.0013	0.0039 ± 0.0008	0.018	0.650
Citramalate	0.4550 ± 0.3264	0.3346 ± 0.1484	0.044	0.550

**Table 5 ijms-26-01282-t005:** Bayesian linear regression. Significant metabolites were found by Bayesian linear regression models with disease (SC > ME/CFS and ME/CFS > SC), gender (male > female), and exercise (non-exercise > post-exercise) as independent variables, age and BMI as covariates, and significance determined from 95% confidence intervals that excluded 0.

Metabolite	ANOVA	Bayes Factor	Disease	Gender	Exercise	Age	BMI
5-Methyltetrahydrofolate	<0.0005	1.04 × 10^9^	SC		Non		
Sarcosine	<0.0005	3365	ME/CFS	Male		Age	Positive
Glucuronate	<0.0005	3348	ME/CFS	Male		Age	
Trans-aconitate	<0.0005	2629	ME/CFS	Male		Age	
7-Methylguanosine	<0.0005	133.7	ME/CFS		Non	Age	
Serine	<0.0005	124.7	ME/CFS			Age	Negative
PG (18:1/18:2)	0.001	50.19	ME/CFS		Non	Age	
SM (d18:1/20:5)	0.001	11.07	ME/CFS			Age	
PG (18:0/18:2)	0.001	9.769	ME/CFS		Non	Age	
SM (d18:1/22:5)	0.001	6.394	ME/CFS			Age	
Glutamine	0.002	0.8696	ME/CFS	Male		Age	
Creatinine	0.004	0.2986	ME/CFS	Male	Non	Age	
Adenosine monophosphate	0.072	0.009596	ME/CFS	Male			
L-Acetylcarnitine	0.085	0.008	ME/CFS				
Creatine	0.203	0.002343	ME/CFS				

**Table 6 ijms-26-01282-t006:** Union of metabolites for disease. Metabolites that were significantly different for the disease factor (ME/CFS vs. SC) in frequentist multivariate analysis (Figure 2) and Bayesian linear regression (Appendix A) were cross-referenced.

Multivariate SC > ME/CFS	Bayesian Regression SC > ME/CFS	Multivariate and Bayesian Regression ME/CFS > SC	Multivariate ME/CFS > SC	Bayesian Regression ME/CFS > SC
1-Methyl-L-Histidine	5-MTHF	Serine	Citric acid	Creatine
Phenylacetyl-L-Glutamine		Sarcosine	Cysteamine	Creatinine
		PG (18:0/18:2)	PE (18:0/22:5)	Trans-aconitate
		PG (18:1/18:2)	PE (18:1/14:1)	L-Acetylcarnitine
		SM (d18:1/20:5)	PE (P-18:0/20:4)	Glutamine
		SM (d18:1/22:5)	PG (18:2/18:2)	Glucuronate
			SM (d18:1/18:4)	AMP
			SM (d18:1/20:0)	7-Methylguanosine
			SM (d18:1/20:4)	
			TAG52:5-FA16:1	

**Table 7 ijms-26-01282-t007:** KEGG pathways for ME/CFS and SC.

SC > ME/CFS	ME/CFS > SC	Combined Analytes
Pyrimidine metabolism	Pyrimidine metabolism	Pyrimidine metabolism
Methionine metabolism	Methionine metabolism	Methionine metabolism
	Pantothenate and CoA biosynthesis	Pantothenate and CoA biosynthesis
	Ammonia recycling	Ammonia recycling
Arginine and proline metabolism		Arginine and proline metabolism
Spermidine and spermine metabolism		Spermidine and spermine biosynthesis
		Glycine and serine metabolism
		Phenylacetate metabolism
		Urea cycle
		Biotin metabolism
Betaine Metabolism		
Riboflavin metabolism		
	Phenylalanine and tyrosine metabolism	
	Taurine and hypotaurine metabolism	
	Starch and sucrose metabolism	

## Data Availability

All data generated or analyzed during this study are included in this published article and its Appendix A. The results of the multivariate statistical analysis are in Appendix A and the log normalized and standardized metabolomics and lipidomics and questionnaire data are in Appendix A.

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
