# Peer review of "Exertional Exhaustion (Post-Exertional Malaise, PEM) Evaluated by the Effects of Exercise on Cerebrospinal Fluid Metabolomics–Lipidomics and Serine Pathway in Myalgic Encephalomyelitis/Chronic Fatigue Syndrome"

_ijms, 2025, doi:10.3390/ijms26031282_

Round 1
Reviewer 1 Report
Comments and Suggestions for Authors
The authors examined metabolic and phospholipid profiles in cerebrospinal fluid between a group of patients with Myalgic Encephalomyelitis / Chronic Fatigue Syndrome and a control group. They found baseline differences between the groups that indicated disease-related pathologies, and in particular between the non-exercising and exercising groups. They used Bayesian methods to gain additional insight into the altered pathways and quantified the effects of age, sex, body mass index, and submaximal exercise. The metabolic and phospholipid profiles suggest the additional hypothesis that white matter dysfunction may contribute to cognitive dysfunction in myalgic encephalomyelitis.The study is well designed, presented, and described. I consider that it can be accepted for publication in the IJMS
Author Response
Thank you for your review and approval of our manuscript.
Reviewer 2 Report
Comments and Suggestions for Authors
“Exertional exhaustion (postexertional malaise, PEM) evaluated by the effects of exercise on cerebrospinal fluid metabolomics, lipidomics and serine pathway in Myalgic Encephalomyelitis/Chronic Fatigue Syndrome.” by James N Baraniuk.
General comments:
This is a highly valuable translational study on a hypothesis about CSF metabolism in ME/CFS. In the present workup, it was revealed that elevated serine and its derivatives sarcosine and phospholipids with a decrease in 5-MTHF, which suggests general dysfunction of folate and one carbon metabolism in ME/CFS. However, there are some concerns regarding publication.
Specific concerns:
The Abstract is sufficient as a letter or expert opinion, however, if the paper will be published as an Original Article, the balanced volume of each section would be needed. Methods and Results should be described more in detail. Similarly, the Introduction is redundant and makes it difficult to understand the specific purpose and significance of this paper. I would like you to focus on the main points of this study and clarify what has been unclear based on the previous articles. For instance, Fig. 4, 5 and 10 are hard to check the details and these figures may be fit for supplemental figures.
Instead, given that it says, “The purpose of this manuscript was to compare the multivariate results to Bayesian analyses,” it would be better to describe the Bayesian analyses in more detail for a wide range of the readers. Also, the authors previously studied the cognitive condition between the Long COVID and ME/CFS. I wonder if the present study also includes ME/CFS related to long COVID or not. If so, it is important to compare and/or discuss these characteristics in this study. In addition, the selection of secondary control of ME/CFS (SC) seems to be quite difficult to find out. In that meaning, it would be meaningful to utilize long COVID group as to compare ME/CFS.
Although only one author is listed, please check again whether all the experiments and writing were done by one person. I would like to be sure that permission was obtained from the colleagues and that this is not a ambiguous study, just to be sure. Without improvement in these points, it will be difficult to properly evaluate this paper, and I hope to be able to check it in detail after the revisions are made.
Reviewer 3 Report
Comments and Suggestions for Authors
This excessively long manuscript refers to metabolite changes (i.e. in the CSF) as reflecting fatigue symptoms in patients. the topic is of certain value, but the authors must learn how to write scientific articles; the excessive length, complicated figures, color contrasts and endless text and references list all impair understanding of this study and the authors must approach an experienced contributor of scientific articles to re-write this manuscript, redraw the figures and shorten the text.
Reviewer 4 Report
Comments and Suggestions for Authors
The study under review will investigate the cerebrospinal fluid (CSF) metabolomic and lipidomic profiles of patients with myalgic encephalomyelitis/chronic fatigue syndrome (ME/CFS) in relation to exertional fatigue, in particular post-exertional malaise (PEM). This research is timely and relevant given the growing recognition of ME/CFS as a major disability with a significant impact on patients' quality of life.
The study employs a robust methodology, including the use of mass spectrometry for metabolomic analysis, which is a key strength as it allows for a comprehensive assessment of biochemical changes. Approval from the Georgetown University Institutional Review Board and adherence to the Declaration of Helsinki underscore the rigorous ethical standards maintained throughout the study. In addition, informed consent was obtained from all participants, further enhancing the integrity of the study.
Nevertheless, several aspects of the paper require further elaboration to improve clarity and strengthen its contributions:
- A more detailed description of the selection criteria for both ME/CFS patients and controls is essential.
- While cerebrospinal fluid is a valuable medium for studying brain-related metabolites, the study may have overlooked the potential insights offered by other biological fluids, such as plasma or urine, which could provide complementary perspectives on metabolic dysregulation in ME/CFS.
- The identification of elevated serine and decreased
5-methyltetrahydrofolate is noteworthy, but the study does not sufficiently explore how these metabolic changes correlate with clinical symptoms such as fatigue or cognitive dysfunction.
- The paper would benefit from additional information on the mass spectrometry techniques used, including the type of mass spectrometer, sample preparation methods and data acquisition parameters.
Addressing these limitations in future research would increase the robustness of the findings and contribute to a more comprehensive understanding of ME/CFS and its metabolic implications.
Round 2
Reviewer 2 Report
Comments and Suggestions for Authors
The author has sufficiently revised their manuscript.
Author Response
Reviewer 2
General comments:
This is a highly valuable translational study on a hypothesis about CSF metabolism in ME/CFS. In the present workup, it was revealed that elevated serine and its derivatives sarcosine and phospholipids with a decrease in 5-MTHF, which suggests general dysfunction of folate and one carbon metabolism in ME/CFS. However, there are some concerns regarding publication.
Specific concerns:
The Abstract is sufficient as a letter or expert opinion, however, if the paper will be published as an Original Article, the balanced volume of each section would be needed. Methods and Results should be described more in detail. Similarly, the Introduction is redundant and makes it difficult to understand the specific purpose and significance of this paper. I would like you to focus on the main points of this study and clarify what has been unclear based on the previous articles. For instance, Fig. 4, 5 and 10 are hard to check the details and these figures may be fit for supplemental figures.
Please see the Marked Up version of the text for my changes in response to your queries.
The Abstract has been extensively revised.
The Introduction has been revised to emphasize the relevance of the metabolomic evaluation and remove redundancy. The section on the sources of cerebrospinal fluid includes new information about the blood brain barrier and separate metabolomic compartments of neurons and specific sets of glial cells that are likely to be novel to ME/CFS investigators and will remind them of the complexity of the brain processes. The review of metabolomics studies in plasma and other bodily fluids is essential to place the current findings into context.
Relevant additions and clarifications have been added to results and methods.
I will contact the editor and copyeditors to get their advice on the font point size and supplemental figures.
Instead, given that it says, “The purpose of this manuscript was to compare the multivariate results to Bayesian analyses,” it would be better to describe the Bayesian analyses in more detail for a wide range of the readers.
The introduction of Bayesian analysis has been rewritten. The full scope of the philosophy of Bayesian analysis is beyond my reach in this article.
Also, the authors previously studied the cognitive condition between the Long COVID and ME/CFS. I wonder if the present study also includes ME/CFS related to long COVID or not. If so, it is important to compare and/or discuss these characteristics in this study. In addition, the selection of secondary control of ME/CFS (SC) seems to be quite difficult to find out. In that meaning, it would be meaningful to utilize long COVID group as to compare ME/CFS.
Long COVID subjects did not have lumbar puncture. I have no data on metabolomics of cerebrospinal fluid in Long COVID.
Sedentary controls were described more completely in the Methods.
Although only one author is listed, please check again whether all the experiments and writing were done by one person. I would like to be sure that permission was obtained from the colleagues and that this is not a ambiguous study, just to be sure.
I conceived the entire suite of experiments, obtained funding, performed the hands on studies, did the statistical analysis and wrote the paper. The metabolomics were performed as described in the Georgetown University Metabolomics Resource. None of their personnel participated in data analysis or manuscript preparation. Other authors participated in other aspects of the larger research project and are credited on their own manuscripts.
What do you mean by an ambiguous study?
Without improvement in these points, it will be difficult to properly evaluate this paper, and I hope to be able to check it in detail after the revisions are made.
Thank you for your comments. I have tried to comply and clarify your points of interest. I have maintained the expanded descriptions and explanations in Introduction and Discussion so that other readers will be able to interpret my findings and arguments within an updated context of brain physiology and thorough review of plasma and other metabolomics studies.
Reviewer 4 Report
Comments and Suggestions for Authors
I disagree with the authors and suggest that, for the sake of future comparability of results, the type of mass spectrometer, sample preparation methods and data acquisition parameters should be specified.
Author Response
Reviewer 4
The study under review will investigate the cerebrospinal fluid (CSF) metabolomic and lipidomic profiles of patients with myalgic encephalomyelitis/chronic fatigue syndrome (ME/CFS) in relation to exertional fatigue, in particular post-exertional malaise (PEM). This research is timely and relevant given the growing recognition of ME/CFS as a major disability with a significant impact on patients' quality of life.
The study employs a robust methodology, including the use of mass spectrometry for metabolomic analysis, which is a key strength as it allows for a comprehensive assessment of biochemical changes. Approval from the Georgetown University Institutional Review Board and adherence to the Declaration of Helsinki underscore the rigorous ethical standards maintained throughout the study. In addition, informed consent was obtained from all participants, further enhancing the integrity of the study.
Nevertheless, several aspects of the paper require further elaboration to improve clarity and strengthen its contributions:
- A more detailed description of the selection criteria for both ME/CFS patients and controls is essential.
The Methods have been amended to emphasize that all ME/CFS and control met the standard exclusions for ME/CFS such as chronic medical disease. ME/CFS subjects met Fukuda and Canadian criteria including postexertional malaise. Sedentary control subjects did not meet even the minimal Fukuda criteria and some may have had individual symptoms such as fatigue without other symptoms (chronic idiopathic fatigue). They may have had mild symptoms but would not meet Fukuda or Canadian criteria. The symptom scores reflect these differences between ME/CFS and SC groups.
- While cerebrospinal fluid is a valuable medium for studying brain-related metabolites, the study may have overlooked the potential insights offered by other biological fluids, such as plasma or urine, which could provide complementary perspectives on metabolic dysregulation in ME/CFS.
Other body fluids will be described separately. Inclusion here would greatly complicate our comparisons and would not be acceptable to the other reviewers.
- The identification of elevated serine and decreased 5-methyltetrahydrofolate is noteworthy, but the study does not sufficiently explore how these metabolic changes correlate with clinical symptoms such as fatigue or cognitive dysfunction.
There was no correlation of serine, 5MTHF or other analyte levels with fatigue, cognition, disability (SF36 domain scores), pain, or other symptoms using Spearman correlation, PCA or other clustering methods. Unfortunately these findings are reported in our companion paper and will not be repeated again in this paper.
- The paper would benefit from additional information on the mass spectrometry techniques used, including the type of mass spectrometer, sample preparation methods and data acquisition parameters.
All methods are described in the text and accompanying citations. The text is taken from the standard operating procedures of the Metabolomics Resource. The reviewer may wish to contact that laboratory for specific requests.
Addressing these limitations in future research would increase the robustness of the findings and contribute to a more comprehensive understanding of ME/CFS and its metabolic implications.
Thank you for your comments. I have tried to revise the manuscript as concisely as possible to match your recommendations.